# *Aspergillus fumigatus* promotes tumor angiogenesis via SLC7A11 on myeloid-derived suppressor cells

Wei Qu [1,2], Zelin Wang[1,2], Tianchen Zhu[1,2], Huiyue Cui[1,2], Ziqian Bing[1,2], Sunan Shen[1,2], Yi Shen [3✉], Shaorong Yu [4✉], Hongqin Zhuang [1✉] & Tingting Wang [1,2✉]

## Abstract

The microbiome is increasingly recognized as playing a critical role in lung cancer prevention, diagnosis, and treatment. While bacteria are essential for tumor angiogenesis, the impact of fungi on this process remains largely unexplored. In this study, we investigate effects of *Aspergillus fumigatus* (*A. fumigatus*) on lung cancer. We show that inhalation of *A. fumigatus* increases tumor burden and angiogenesis in mouse models. Interestingly, *A. fumigatus* does not directly affect the proangiogenic abilities of tumor cells or endothelial cells. Instead, *A. fumigatus* promotes the accumulation of myeloid-derived suppressor cells (MDSCs), particularly G-MDSCs, in tumor tissues. *A. fumigatus* increases VEGF-A secretion from tumor-associated MDSCs, promoting tumor angiogenesis. Furthermore, we identify solute carrier family 7 member 11 (SLC7A11) as a key player in regulating this proangiogenic function through an interaction with High Mobility Group Box 1 (HMGB1) in MDSCs. Our results shed light on the mechanisms by which *A. fumigatus* influences MDSCs to promote angiogenesis and demonstrate that commensal fungi influence host immunity and support tumor progression.

**Keywords** *Aspergillus fumigatus*; Angiogenesis; Lung Cancer; MDSCs
**Subject Categories** Cancer; Microbiology, Virology & Host Pathogen Interaction; Vascular Biology & Angiogenesis

## Introduction

Microbes have been implicated in various types of cancer and play a crucial role in tumor diagnosis, prognosis, treatment, and recurrence (Sepich-Poore et al, 2021). Commensal microorganisms are linked to the immune response to tumors, influenced by factors in both the tumor microenvironment and the systemic periphery (Buchta Rosean and Rutkowski, 2017). For example, in pancreatic cancer, fungi or fungal products can stimulate the secretion of

interleukin-33 (IL-33) from PDAC cells, leading to a type II immune response that supports tumor growth by creating an immunosuppressive environment (Alam et al, 2022). Our previous research has shown that intestinal fungi can drive the malignant progression of colorectal tumors by modulating the differentiation and immunoregulatory functions of MDSCs in the tumor microenvironment (Wang et al, 2018).

Research on the role of microorganisms in tumor regulation is primarily focused on intestinal tumors, with limited reports on the relationship between lung disease and the lung microbiota. Additionally, emphasis is placed on the role of bacteria, such as symbiotic bacteria in the lungs activating γδ T cells to promote lung cancer (Jin et al, 2019). *Veillonella parvula* was found to modulate host immune responses in the lower airways, leading to reduced survival, increased tumor burden, an interleukin-17 (IL-17) inflammatory phenotype, and the activation of checkpoint inhibitor markers in KP lung cancer mice (Tsay et al, 2021). There is a notable research gap regarding the relationship between dysbiosis of fungi and specific oncogenic effects (Kaplun et al, 2021; Kaźmierczak-Siedlecka et al, 2020; Saftien et al, 2023). Thus, targeting the link between pulmonary fungi and lung cancer development could offer new therapeutic approaches for cancer prevention.

Tumor angiogenesis is a key process in cancer progression (Vimalraj, 2022; Zhou et al, 2022). Recent studies suggest a connection between angiogenic processes and the microbiota in various human diseases (Sajib et al, 2018). For instance, colonizing germ-free mice with *P. micra* strains from colorectal cancer patients has been shown to upregulate genes linked to angiogenesis in the colon tissue (Zhao et al, 2022). Additionally, *P. gingivalis* was found to promote tumor progression in oral cancer models by enhancing angiogenesis (Lamont et al, 2022). Mechanistically, epithelial cells secrete interleukin-1β (IL-1β) in response to *P. gingivalis* stimulation. This activation subsequently induces endothelial cells to exert pro-angiogenic effects. While bacterial involvement in angiogenesis is known, the effects of fungi on tumor angiogenesis remain unexplored.

MDSCs are immature bone marrow-derived immune cells that can suppress the immune function of the organism. In conditions such as cancer and chronic infections, MDSCs have

[1]State Key Laboratory of Pharmaceutical Biotechnology, Medical School & School of Life Sciences, Nanjing University, Nanjing 210093, China. [2]Jiangsu Key Laboratory of Molecular Medicine, Division of Immunology, Medical School, Nanjing University, Nanjing 210093, China. [3]Department of Cardiothoracic Surgery, Jinling Hospital, Medical School of Nanjing University, 305 East Zhongshan Road, Nanjing, China. [4]Jiangsu Cancer Hospital, Jiangsu Institute of Cancer Research, Nanjing Medical University Affiliated Cancer Hospital, Nanjing 210009 Jiangsu Province, China. ✉E-mail: dryishen@nju.edu.cn; shaorongyu@njmu.edu.cn; hqzhuang@nju.edu.cn; wangtt@nju.edu.cn

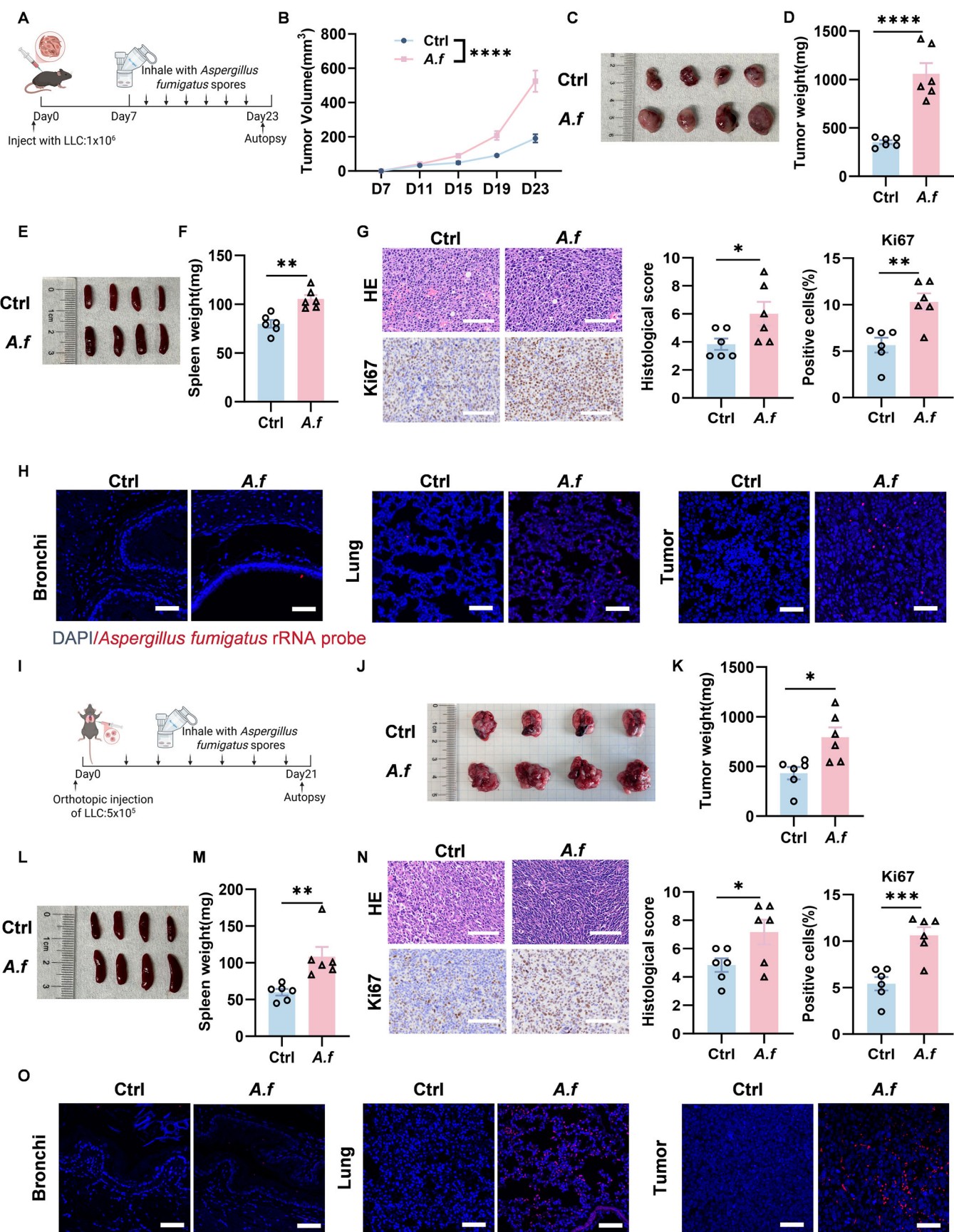

**Figure 1. A. fumigatus promotes development of lung cancer.**

(A) Lewis lung cancer cells (1 × 10⁶) were injected subcutaneously into the right abdomen of mice. Every 2 days, mice inhaled live *A. fumigatus* spores. All tumor-bearing mice were executed on day 23 (Created with BioRender.com.). (B) Tumor-volume curve was calculated after injection every four days ($p < 0.0001$) ($n = 6$ biological replicates) (C) Representative images of tumors from different groups. (D) Tumors isolated from mice were weighed ($p < 0.0001$) ($n = 6$ biological replicates). (E) Representative images of spleens from different groups. (F) Spleens isolated from mice were weighed ($p = 0.0011$) ($n = 6$ biological replicates). (G) Histological analysis of tumors was shown by HE staining (scale bars, 100 μm) ($p = 0.0449$). Immunohistochemical analysis of Ki67 in tumors (scale bars, 100 μm) ($p = 0.0032$) ($n = 6$ biological replicates). (H) Detection of *A. fumigatus* in mouse bronchial, lung, and tumor tissues by fluorescence in situ hybridization (FISH) (scale bars, 50 μm). (I) Mice were allowed to inhale live *A. fumigatus* spores every 2 days based on an orthotopic lung cancer mouse model. All tumor-bearing mice were executed on day 21 (Created with BioRender.com.). (J) Representative images of tumors from different groups. (K) Tumors isolated from mice were weighed ($p = 0.0112$) ($n = 6$ biological replicates). (L) Representative images of spleens from different groups. (M) Spleens isolated from mice were weighed ($p = 0.0060$) ($n = 6$ biological replicates). (N) Histological analysis of tumors was shown by HE staining (scale bars, 100 μm) ($p = 0.0409$). Immunohistochemical analysis of Ki67 in tumors (scale bars, 100 μm) ($p = 0.0009$) ($n = 6$ biological replicates). (O) Detection of *A. fumigatus* in mouse bronchial, lung, and tumor tissues by FISH (scale bars, 50 μm). Data information: Data with error bars are represented as mean ± SEM. Tumor growth curves were analyzed by two-way ANOVA. Other data were analyzed using unpaired Student's t-test. *$p < 0.05$, **$p < 0.01$, ***$p < 0.001$, ****$p < 0.0001$ as determined by unpaired Student's t-test or two-way ANOVA. Source data are available online for this figure.

been shown to modify the immune response and facilitate tumor development (He et al, 2018). Studies have connected MDSCs to yeast-induced tumor progression in colorectal and lung cancers (Liu et al, 2023; Wang et al, 2018), although the exact mechanisms of this relationship have not been thoroughly investigated.

In this study, we examine the impact of *A. fumigatus* on the progression of lung cancer, revealing that the fungus promotes angiogenesis in tumors through MDSCs. *A. fumigatus* burden leads to an accumulation of MDSCs, particularly G-MDSCs, within tumors. Reduction of G-MDSCs mitigates enhanced tumor angiogenesis induced by peritumoral *A. fumigatus* treatment. We also find that *A. fumigatus* modulates pro-angiogenic effects by influencing the interaction between SLC7A11 and HMGB1 in MDSCs. Targeting this interaction could potentially reduce tumor angiogenesis observed in mice treated with *A. fumigatus*.

## Results

### A. fumigatus promotes the development of lung cancer and tumor angiogenesis

Dysbiosis of the lung microbiome has been associated with the development of lung disease (Mao et al, 2018). Haziza et al. calculated the intratumoral fungal abundance in the TCGA cohort (Narunsky-Haziza et al, 2022). Using the decontaminated TCGA fungal count data (Narunsky-Haziza et al, 2022), we analyzed the abundance of different fungal genera in Lung Adenocarcinoma (LUAD) and found that *Aspergillus* was significantly more abundant than others (Fig. EV1A). In the clinic, fungi such as *A. fumigatus* were found at the tumor site in lung cancer patients (Bao et al, 2019; Bassetti and Bouza, 2017). In human lung adenocarcinoma tumor samples, we found that *A. fumigatus* localized in the tumor tissue by fluorescence in situ hybridization (FISH) experiments (Fig. EV1B). To investigate the role of *A. fumigatus* in lung cancer, we established a Lewis lung adenocarcinoma model in mice and performed inhalation experiments with live *A. fumigatus* spores (Fig. 1A). Our findings revealed that mice in the *A. fumigatus*-treated group experienced faster tumor growth (Fig. 1B–D). Furthermore, splenomegaly was more severe in the *A. fumigatus*-treated group compared to the control group (Fig. 1E,F). The results of HE staining and

immunohistochemical Ki67, an indicator of tumor proliferation, demonstrated that *A. fumigatus* led to increased tumor progression (Fig. 1G). Furthermore, we found that *A. fumigatus* localized predominantly in tumor tissues and lung tissues of mice by FISH experiments (Fig. 1H). Besides, to simulate lung microbiota dysbiosis, we established an orthotopic lung cancer model and made mice inhale live *A. fumigatus* spores (Fig. 1I). *A. fumigatus* accelerated tumor growth (Fig. 1J,K), exacerbated splenomegaly (Fig. 1L,M). The results of HE staining and immunohistochemical Ki67 indicated that *A. fumigatus* also led to exacerbation of tumor progression based on the orthotopic model of lung cancer (Fig. 1N). Similarly, we found that *A. fumigatus* localized predominantly to tumor and lung tissues in an orthotopic lung cancer model (Fig. 1O). To verify the credibility of our conclusions, we also injected live *A. fumigatus* peritumorally in LLC model mice (Fig. EV1C). We found that *A. fumigatus* also promoted tumor growth (Fig. EV1D–F), splenomegaly (Fig. EV1G,H), and tumor progression in mice (Fig. EV1I). These findings confirmed that *A. fumigatus* promotes the development of lung cancer.

Subsequently, we identified the pathways highly enriched in *A. fumigatus*-positive samples by Gene Set Enrichment Analysis (GSEA) with a threshold of adjusted *P*-value < 0.05, based on the results of Narunsky-Haziza et al (Narunsky-Haziza et al, 2022). We found that gene expression of angiogenesis-related pathways was significantly upregulated (Fig. 2A). Therefore, we explored whether tumor angiogenesis was increased in the *A. fumigatus*-treated group. We first performed assays in LLC mouse model with inhalation of live *A. fumigatus* spores. We observed increase expression of angiogenesis markers CD34 and CD31 (Fig. 2B,C) and a significant upregulation of *Vegf-a* gene (Fig. 2D) in tumor tissues. Concurrently, VEGF-A protein levels were increased in tumor tissue and serum (Fig. 2C,E). In addition, *A. fumigatus* spore inhalation elevated CD34 and CD31 (Fig. 2F,G), upregulated *Vegf-a* gene levels (Fig. 2H), and increased VEGF-A in tumors and serum (Fig. 2I,J) in orthotopic lung cancer models. These results suggested that *A. fumigatus* exacerbates angiogenesis in the lung adenocarcinoma model.

### A. fumigatus promotes pro-angiogenic function of MDSCs

Next, we explored the target cells of *A. fumigatus* in lung cancer. We used the HUVEC endothelial cell line to simulate vascular

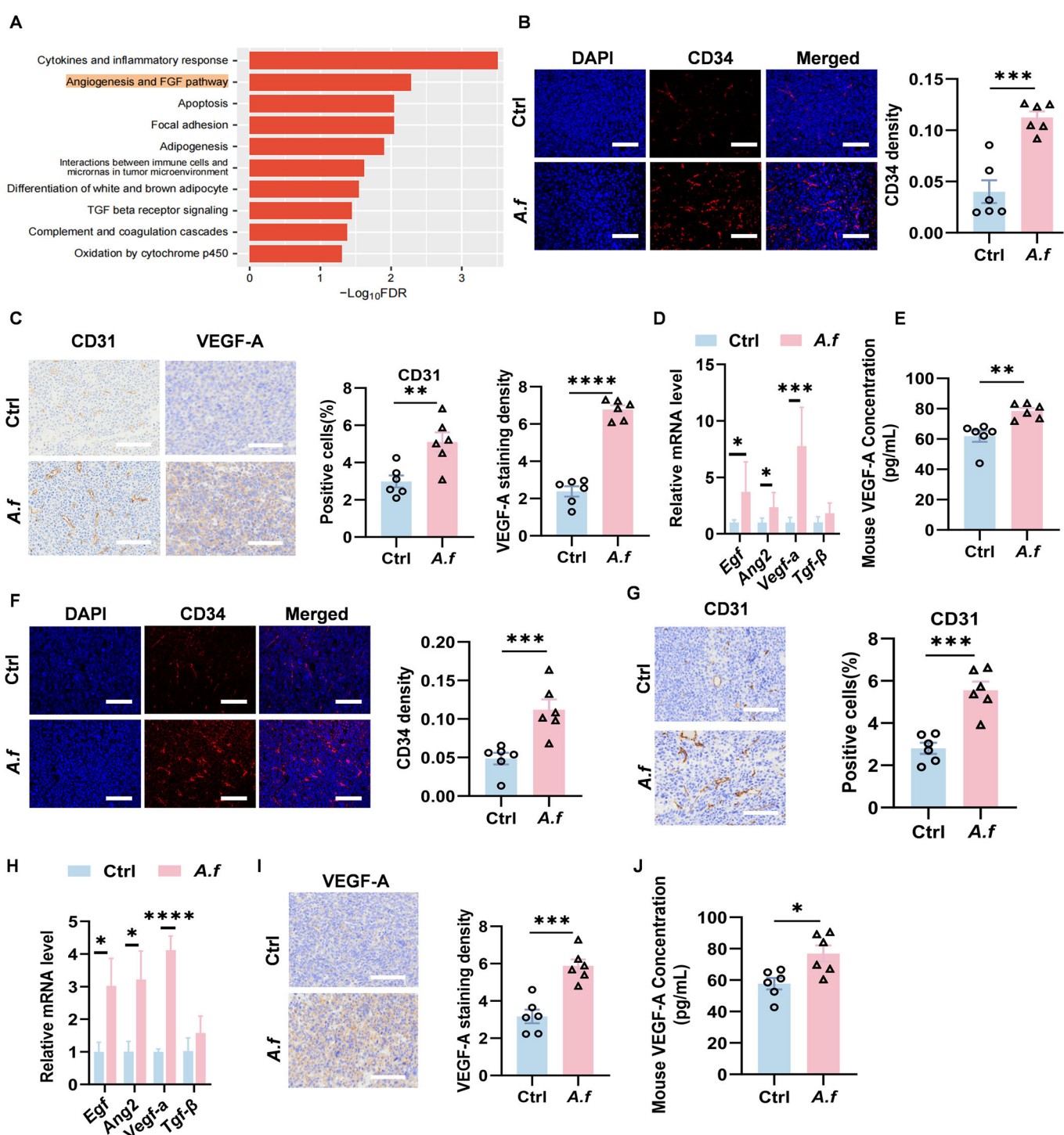

endothelial cells (Cao et al, 2017). We found that *A. fumigatus* did not significantly affect endothelial cell migration, proliferation, expression of angiogenesis related genes, secretion of VEGF-A or lumen formation (Fig. EV2A–F). To identify the interaction between fungi and lung cancer cells, we co-cultured A549 lung adenocarcinoma cells with *A. fumigatus*. The results showed no significant impact on A549 cell migration, invasion, proliferation,

or angiogenic capacity (Fig. EV2G–L). Consistent with other cell types, *A. fumigatus* did not alter migration, proliferation, or angiogenesis-related gene expression in human SPCA1 (Fig. EV2M–O) or murine LLC lung cancer cells (Fig. EV2P–R).

The tumor possesses a distinct microenvironment that can influence surrounding cells, significantly impacting tumor progression (Genova et al, 2021; Pitt et al, 2016). Therefore, we

**Figure 2.** *A. fumigatus* promotes tumor angiogenesis in lung cancer.

(A) Angiogenesis and FGF signaling pathways highly enriched in *A. fumigatus*-positive TCGA tumor samples identified by GSEA. (B) Immunofluorescence images of tumor tissue CD34 in mouse LLC model (scale bars, 100 μm) ($p = 0.0002$) ($n = 6$ biological replicates). (C) Immunohistochemical analysis of CD31 and VEGF-A in tumors (scale bars, 100 μm) (CD31: $p = 0.0063$; VEGF-A: $p < 0.0001$) ($n = 6$ biological replicates). (D) The gene expression related to angiogenesis in mouse LLC model was detected by qPCR. (Relative mRNA level of *Egf*: $p = 0.0316$; Relative mRNA level of *Ang2*: $p = 0.0360$; Relative mRNA level of *Vegf-a*: $p = 0.0007$) ($n = 6$ biological replicates). (E) VEGF-A in serum was assessed ($p = 0.0028$) ($n = 6$ biological replicates). (F) Immunofluorescence images of tumor tissue CD34 in orthotopic lung cancer model (scale bars, 100 μm) ($p = 0.002$) ($n = 6$ biological replicates). (G) Immunohistochemical analysis of CD31 in tumors (scale bars, 100 μm) ($p = 0.0002$) ($n = 6$ biological replicates). (H) The gene expression related to angiogenesis in orthotopic lung cancer model was detected by qPCR. (Relative mRNA level of *Egf*: $p = 0.0471$; Relative mRNA level of *Ang2*: $p = 0.0376$; Relative mRNA level of *Vegf-a*: $p < 0.0001$) ($n = 6$ biological replicates). (I) Immunohistochemical analysis of VEGF-A in tumors (scale bars, 100 μm) ($p = 0.0003$) ($n = 6$ biological replicates). (J) VEGF-A in serum was assessed ($p = 0.0128$) ($n = 6$ biological replicates). Data information: Data with error bars are represented as mean ± SEM. Data were analyzed using unpaired Student's t-test. *$p < 0.05$, **$p < 0.01$, ***$p < 0.001$, ****$p < 0.0001$ as determined by unpaired Student's t-test. Source data are available online for this figure.

investigated whether *A. fumigatus* induces angiogenesis by modulating the tumor microenvironment. We quantified immune cell infiltration in tumor tissues from the LLC mouse model (Fig. 3A). The results of flow experiments showed that *A. fumigatus* treatment did not significantly alter the proportions of macrophages, dendritic cells, Treg cells and CD4$^+$ T cells in the tumor microenvironment (Fig. EV3A–D). Notably, there is a significantly increase in the number of MDSCs within mouse tumors upon *A. fumigatus* treatment, predominantly consisting of G-MDSCs (Fig. 3B,C). The number of MDSCs in the bone marrow and spleen did not change significantly (Fig. EV3E,F). Thus, the effect of *A. fumigatus* appears to be largely confined to the tumor site, with minimal impact on the bone marrow and spleen. Moreover, we observed a concomitant reduction in CD8$^+$ T cells infiltration within tumors (Fig. EV3G). Considering that MDSCs increase in the tumor microenvironment in the *A. fumigatus*-treated group, we investigated whether *A. fumigatus* could induce proangiogenic effects through MDSCs. To validate the involvement of G-MDSCs in tumor angiogenesis, we also isolate the LY6G$^+$ cells from the *A. fumigatus*-treated group in tumor-bearing mice using the method described by He J et al (He et al, 2025) (Fig. 3A). The supernatants of LY6G$^+$ cells were collected after 48 h. Compared with the control group, the supernatant of the *A. fumigatus*-treated group promoted the formation of the endothelial cell lumen and showed higher VEGF-A levels (Fig. 3D,E).

To validate the role of MDSCs in *A. fumigatus*-accelerated tumor progression, bone marrow-derived MDSCs were treated with *A. fumigatus* and analyzed by flow cytometry 24 h later (Fig. 3F). The results showed that *A. fumigatus* could also induce increased differentiation of MDSCs in vitro, with a greater bias towards G-MDSCs (Fig. 3G,H). We collected the cell culture supernatant from MDSCs as the conditioned medium (Fig. 3F). The conditioned medium from MDSCs significantly promoted the lumen formation of endothelial cells (Fig. 3I). We found that the concentration of VEGF-A in the supernatant of MDSCs was higher in the *A. fumigatus* group (Fig. 3J). Functional validation of VEGF-A involvement was performed. VEGF-A knockdown via siRNA in MDSCs decreased angiogenesis induction by *A. fumigatus* (Figs. 3K and EV3H). Moreover, anti-VEGF-A antibody neutralization suppressed endothelial lumen formation promoted by conditioned media from *A. fumigatus* treated MDSCs (Fig. 3L). Furthermore, we confirmed that *A. fumigatus* predominantly co-localized with the myeloid cell marker CD11b, but not the tumor cell marker panCK, as

well as the endothelial cell marker CD31 in tumor tissues by FISH experiments (Fig. EV3I–K). In conclusion, we demonstrate that *A. fumigatus* can promote angiogenesis in lung cancer through MDSCs.

## G-MDSCs increase tumor burden in tumor-bearing mice peritumorally injected with *A. fumigatus*

In order to verify the tumor-promoting role of G-MDSCs, we used anti-Ly6G antibody to reduce G-MDSCs in mice (Fig. 4A). Although the anti-Ly6G treatment did not affect the proportion of CD4$^+$ T cell subsets, there was a rebound in the number of CD8$^+$ T cells (Fig. EV4A). Besides, the increase in the number of MDSCs and G-MDSC was suppressed (Fig. EV4B,C). In the mouse Lewis Lung Carcinoma (LLC) model, following G-MDSCs reduction, the exacerbated tumor burden caused by *A. fumigatus* was alleviated, and the tumor growth rate was decelerated (Fig. 4B–D). Splenomegaly was relieved (Fig. 4E,F). The percentage of Ki67 positive cells in tumor tissues and the results of the HE staining showed a decrease in tumor progression (Figs. 4G and EV4D). In addition, angiogenesis-related indicators were also reduced in the G-MDSCs reduction group (Fig. 4H–L).

## SLC7A11 mediates the pro-angiogenic function of MDSCs induced by *A. fumigatus*

To investigate how *A. fumigatus* influences the function of MDSCs, we examined differential gene expression in MDSCs. We observed upregulation of glutamine metabolism (Fig. 5A,B). Subsequently, our study identified SLC7A11 as the most upregulated protein in the glutamine pathway (Fig. 5C). SLC7A11 is a solute carrier responsible for the reverse transport of cystine and glutamate (Jyotsana et al, 2022). Recent studies have shown that SLC7A11 is overexpressed in various cancer types and is associated with poor patient prognosis (Liu et al, 2020; Shen et al, 2022; Zhang et al, 2021). Therefore, SLC7A11 may serve as a novel therapeutic target for cancer treatment. Our experimental results also showed that the SLC7A11-specific inhibitor sulfasalazine (SASP) was able to inhibit the increase in proangiogenic function induced by *A. fumigatus* in MDSCs (Figs. 5D,E and EV5A,B).

In vivo studies confirmed that SASP attenuated *A. fumigatus*-aggravated lung cancer progression (Fig. 5F). SASP significantly suppressed *A. fumigatus*-induced increase in tumor mass and volume (Fig. 5G–I) and ameliorated splenomegaly (Fig. 5J,K).

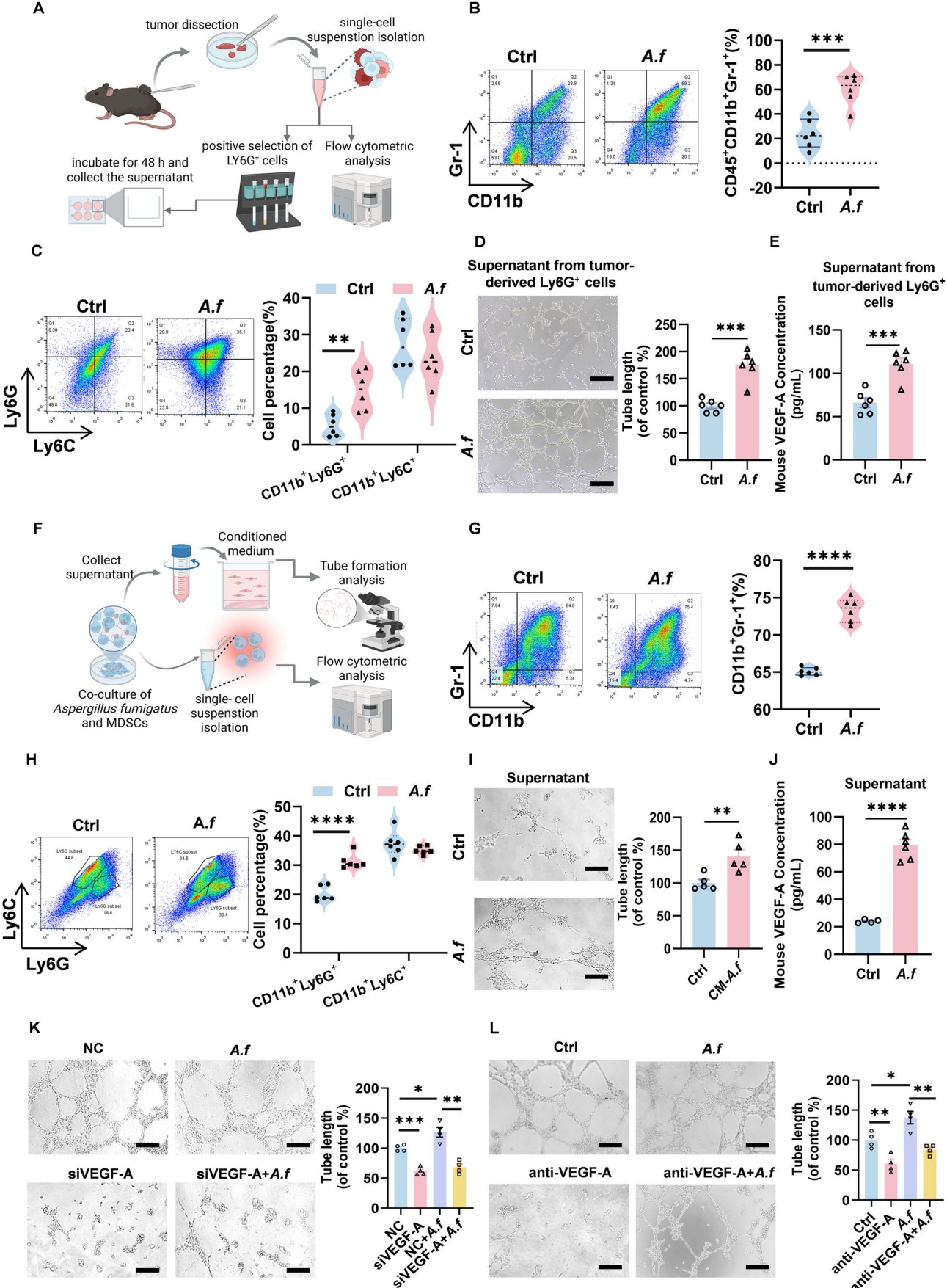

**Figure 3.** ***A. fumigatus* promotes the pro-angiogenic function of MDSCs.**

(A) In the LLC model with peritumoral injection of *A. fumigatus*, tumor tissue was collected from mice and processed into single-cell suspensions, which were then analyzed by flow cytometry. Separately, LY6G⁺ cells were isolated from mice by magnetic-activated cell sorting. The isolated LY6G⁺ cells were cultured for 48 h, and the supernatant was collected (Created with BioRender.com.). (B, C) Proportions of total MDSCs (CD45⁺CD11b⁺Gr-1⁺) ($p = 0.0005$), G-MDSCs (CD11b⁺Ly6G⁺) ($p = 0.0037$) and M-MDSCs (CD11b⁺Ly6C⁺) in tumor tissues were detected by flow cytometry ($n = 6$ biological replicates). (D) HUVECs were stimulated with supernatant from cultured tumor-derived LY6G⁺ cells. Tube formation was then assessed (scale bars, 200 μm). ($p = 0.0001$) ($n = 6$ biological replicates). (E) The level of VEGF-A in the supernatant of cultured tumor-derived LY6G⁺ cells was measured by ELISA ($p = 0.0004$) ($n = 6$ biological replicates). (F) MDSCs were stimulated with heat-inactivated *A. fumigatus* (MOI = 2) for 24 h. The supernatant was collected. The cells were prepared into single-cell suspensions (Created with BioRender.com.). (G, H) Proportions of total MDSCs ($p < 0.0001$), G-MDSCs ($p < 0.0001$) and M-MDSCs were detected by flow cytometry ($n = 6$ biological replicates). (I) The HUVEC cells were stimulated with the co-culture supernatant. The lumen-forming capacity of HUVEC cells was assessed (scale bars, 200 μm) ($p = 0.0081$) ($n = 4$ biological replicates). (J) ELISA analysis of VEGF-A secretion in the culture supernatant of MDSCs stimulated with *A. fumigatus*. ($p < 0.0001$) ($n = 4$ biological replicates). (K) MDSCs were transfected with VEGF-A siRNA for 48 h before stimulation with heat-inactivated *A. fumigatus* (MOI = 2) for 24 h. The HUVEC cells were stimulated with the co-culture supernatant. The lumen-forming capacity of HUVEC cells were assessed (scale bars, 200 μm) (NC and siVEGF-A: $p = 0.0001$; NC and NC + A. f: $p = 0.0232$; NC + A. f and siVEGF-A + A. f: $p = 0.001$) ($n = 4$ biological replicates). (L) The supernatant was collected from MDSCs stimulated with heat-inactivated *A. fumigatus* (MOI = 2) for 24 h. The supernatant was preincubated with anti-VEGF-A-neutralizing antibody or with the isotype control. The lumen-forming capacity of supernatant-stimulated HUVEC cells was evaluated (scale bars, 200 μm). (Ctrl and anti-VEGF-A: $p = 0.0073$; Ctrl and A. f: $p = 0.0223$; Ctrl+A. f and anti-VEGF-A + A. f: $p = 0.0031$) ($n = 4$ biological replicates). Data information: Data with error bars are represented as mean ± SEM. Data were analyzed using unpaired Student's t-test. \**p* < 0.05, \*\**p* < 0.01, \*\*\**p* < 0.001, \*\*\*\**p* < 0.0001 as determined by unpaired Student's t-test. Source data are available online for this figure.

The results of the HE staining experiment and the reduced presence of Ki67 positive cells in tumor tissue showed slower tumor progression (Figs. EV5C and 5L). Decreased tumor angiogenesis was demonstrated by downregulated expression of pro-angiogenic markers VEGF-A, CD31 and CD34 (Fig. 5M–O). Concordantly, SASP reduced both serum VEGF-A levels and intratumoral *vegf-a* expression elevated by *A. fumigatus* (Fig. 5P,Q). Collectively, the above results suggest that SASP alleviates accelerated tumor growth and increased angiogenesis caused by *A. fumigatus*.

## HMGB1 interacts with SLC7A11 in MDSCs

To further investigate how SLC7A11 regulates the proangiogenic ability of MDSCs, we purified the proteins that bind to SLC7A11 in MDSCs stimulated with *A. fumigatus* by COIP/MS, using unstimulated MDSCs as control. We focused on angiogenesis-related proteins and found that only HMGB1 (accession ID: P63158) could bind to SLC7A11. The list of a portion of the identified proteins is shown (Fig. 6A). Immunofluorescence showed that SLC7A11 and HMGB1 were colocalized in MDSCs (Fig. 6B). The docking fraction of the SLC7A11 protein to the HMGB1 protein is 1676.447, with 8 hydrogen bonds on the predicted binding surface of the two proteins (Fig. 6C). We used a specific inhibitor of HMGB1, glycyrrhizin. Our experimental results show that glycyrrhizin hinders the co-localization of HMGB1 with SLC7A11 in MDSCs (Fig. 6B). However, glycyrrhizin did not affect the increased G-MDSCs differentiation induced by *A. fumigatus* (Fig. 6D). Furthermore, glycyrrhizin inhibited the *A. fumigatus*-induced increase in VEGF-A content in MDSCs at both gene and protein levels (Fig. 6E,F). The results of lumen formation experiments showed that glycyrrhizin caused a decrease in the proangiogenic function of the conditioned medium of MDSCs (Fig. 6G). Co-immunoprecipitation (CoIP) analysis revealed SLC7A11-HMGB1 interaction (Fig. 6H). Using ZDOCK-predicted binding residues, we generated SLC7A11 point mutants (Asp386, Gln71, Leu299, Phe437, Tyr444). Mutant plasmids were tested via COIP. Asp386, Leu299, and Gln71 mutations significantly weakened HMGB1 binding, while Phe437 and Tyr444

mutations showed no significant effect (Fig. 6I). This validates key binding sites and strengthens our mechanistic model. In summary, SLC7A11 can regulate the pro-angiogenic capacity of MDSCs through an interaction with HMGB1.

In summary, we put forward the following working model: *A. fumigatus* displays pro-angiogenic ability, which leads to the progression of lung cancer. Furthermore, the pro-angiogenic capacity of *A. fumigatus* does not directly affect tumor cells and endothelial cells but acts directly on MDSCs, especially G-MDSCs. *A. fumigatus* induces VEGF-A production in MDSCs, which is controlled by the interaction between SLC7A11 and HMGB1.

## Discussion

The crucial role of angiogenesis in tumor growth and metastasis promotion is well established, and increasing evidence suggests that microorganisms impact tumor therapy outcomes (Wong-Rolle et al, 2021). Colonisation of microorganisms in the intestines of germ-free mice significantly increased intestinal angiogenesis and colonic vessel diameter (Hoffmann et al, 2016; Stappenbeck et al, 2002). Infection of crypt-like intestinal epithelial cells with diffusely adherent *E. coli* increases the expression of HIF-1α protein, leading to upregulation of VEGF-A expression and increased angiogenesis (Cane et al, 2010). In lung tissues, microbiota-induced Th17 cells have been demonstrated to promote tumor angiogenesis (Chang et al, 2014). Furthermore, *C. albicans* infection in the cornea has been shown to stimulate ocular angiogenesis through MMP13, potentially resulting in vision impairment or blindness (Gao et al, 2017). However, the effects of fungi in tumor angiogenesis remain unknown. Recently, fungi have been found widely present in 35 tumors (Narunsky-Haziza et al, 2022). Pan-cancer analyses of multiple body sites have shown that there is 1 tumor-associated fungal biota for every $10^4$ tumor cells (Dohlman et al, 2022). Furthermore, clinical observations show that *Aspergillus* infections and lung cancer can coexist. Cases of lung cancer patients with co-existing pulmonary aspergillosis have been reported (Nilsson et al, 2013). Moreover, *Aspergillus* colonization has been found within endobronchial tumors (Kaplun et al, 2021). These findings suggest

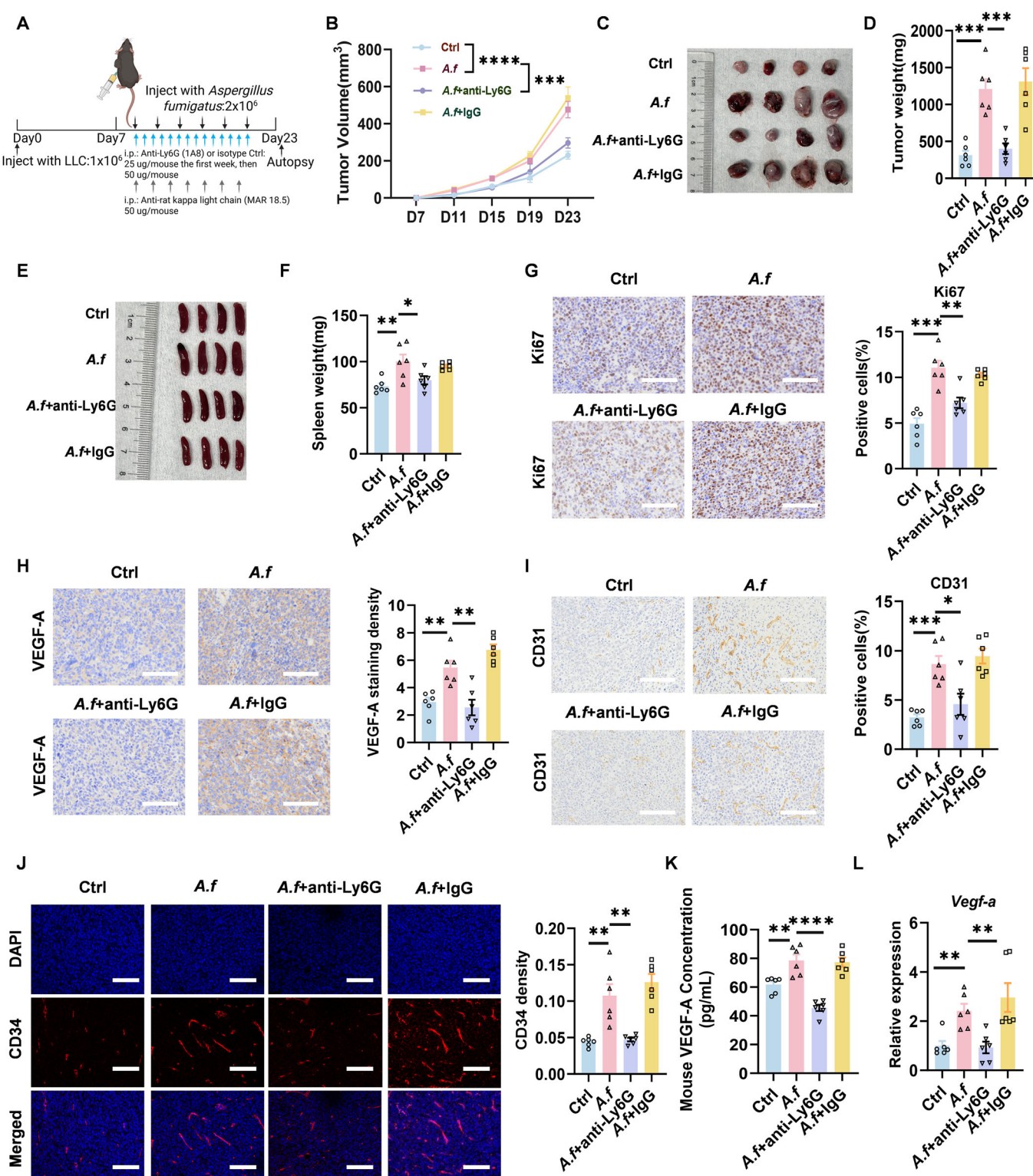

that chronic pulmonary *Aspergillus* infection or colonization may potentially link to lung cancer development. Here, our results revealed that peritumoral treatment with *A. fumigatus* accelerates tumor progression and enhances tumor angiogenesis. Elevated serum VEGF-A levels and upregulated expression of VEGF-A,

CD34 and CD31 markers in tumor tissues confirm the ability of *A. fumigatus* to promote angiogenesis in lung cancer.

The pro-angiogenic effects of microorganisms are mainly considered to act on endothelial cells or tumor cells. Intestinal bacteria can promote angiogenic responses on human intestinal

◄ **Figure 4.  Targeting G-MDSCs suppresses *A. fumigatus*-mediated tumor growth and angiogenesis.**

(A) Tumor-bearing mice were treated for 2 weeks using a combination strategy (Combo) to reduce G-MDSCs according to Boivin G et al (Boivin et al, 2020) (Created with BioRender.com.). (B) Tumor volume were assessed. (Ctrl and *A. f*: $p < 0.0001$; *A. f* and *A. f* + antiLy6G: $p = 0.0002$) ($n = 6$ biological replicates). (C) Representative picture of tumors isolated from different groups. (D) Tumor mass were assessed. (Ctrl and *A. f*: $p = 0.0001$; *A. f* and *A. f* + antiLy6G: $p = 0.0004$) ($n = 6$ biological replicates). (E) Representative picture of spleens. (F) Spleen mass was assessed. (Ctrl and *A. f*: $p = 0.0079$; *A. f* and *A. f* + antiLy6G: $p = 0.0416$) ($n = 6$ biological replicates). (G–I) Tumor tissues were stained for Ki67, VEGF-A, CD31 (scale bars, 100 μm). (Ki67: Ctrl and *A. f*: $p = 0.0001$; *A. f* and *A. f* + antiLy6G: $p = 0.0027$; VEGF-A: Ctrl and *A. f*: $p = 0.0019$; *A. f* and *A. f* + antiLy6G: $p = 0.0034$; CD31: Ctrl and *A. f*: $p = 0.0001$; *A. f* and *A. f* + antiLy6G: $p = 0.0133$) ($n = 6$ biological replicates). (J) Immunofluorescence images of CD34 in tumors (scale bars, 100 μm). (Ctrl and *A. f*: $p = 0.0025$. *A. f* and *A. f* + antiLy6G: $p = 0.0035$) ($n = 6$ biological replicates). (K) VEGF-A in serum was assessed. (Ctrl and *A. f*: $p = 0.0051$; *A. f* and *A. f* + antiLy6G: $p < 0.0001$) ($n = 6$ biological replicates). (L) *Vegf-a* gene expression in tumor tissues was measured by qPCR. (Ctrl and *A. f*: $p = 0.0021$. *A f* and *A. f* + antiLy6G: $p = 0.0026$) ($n = 6$ biological replicates). Data information: Data with error bars are represented as mean ± SEM. Tumor growth curves were analyzed by two-way ANOVA. Other data were analyzed using unpaired Student's t-test. *$p < 0.05$, **$p < 0.01$, ***$p < 0.001$, ****$p < 0.0001$ as determined by unpaired Student's t-test or two-way ANOVA. Source data are available online for this figure.

microvascular endothelial cells (HIMEC) and human intestinal fibroblasts through ligands, such as toll-like receptors (TLRs) and NOD-like receptors (NLRs)(Schirbel et al, 2013). *H. pylori* VacA toxin can increase vascular endothelial growth factor expression in MKN28 gastric cells through an epidermal growth factor receptor and COX2-dependent mechanism (Caputo et al, 2003). Our study found that *A. fumigatus* led to the accumulation of more MDSCs around the tumor, predominantly G-MDSCs. This finding is consistent with previous reports that *Candida tropicalis* stimulation resulted in increased differentiation of G-MDSCs (Wang et al, 2018). Although the study of Ben-Ami R et al. showed that *A. fumigatus* metabolite gliotoxin can inhibit angiogenesis (Ben-Ami et al, 2013; Ben-Ami et al, 2009). We believe that the effects of *A. fumigatus* itself on endothelial cells cannot be ignored compared to the metabolites. We co-cultured *A. fumigatus* directly with endothelial cells and confirmed that *A. fumigatus* had no significant effect on endothelial cell migration, proliferation, and angiogenesis. Ben-Ami R et al. showed that in invasive pulmonary aspergillosis, the *A. fumigatus* metabolite gliotoxin inhibited angiogenesis in a dose-dependent manner. We believe this may be because the large increase in *A. fumigatus* in invasive pulmonary aspergillosis. Therefore, *A. fumigatus* produces large amounts of gliotoxins that directly damage endothelial cells. However, in tumor models, *A. fumigatus* does not increase in large numbers. In the tumor model, *A. fumigatus* affected tumor angiogenesis by regulating MDSCs in the tumor microenvironment. This suggests that there may be differences in angiogenesis between different disease models and *A. fumigatus* amounts.

Additionally, MDSCs are known to have a significant impact on tumor angiogenesis (Rahma and Hodi, 2019). The progression of lung cancer is intricately related to the immune microenvironment (Altorki et al, 2019). Clinical data has shown that MDSCs are highly expressed in intrinsic immune cells of lung adenocarcinoma patients, with a content of 23% (Zhang et al, 2019). We subsequently verified in vitro experiments that *A. fumigatus* could induce increased secretion of VEGF-A in MDSCs and enhance the lumen-forming ability as well as the migratory capacity of HUVEC. Our study demonstrated that *A. fumigatus* promotes angiogenesis and accelerates disease progression by regulating MDSCs in the tumor microenvironment. Notably, Sohrabi et al reported that intravenous injection of *A. fumigatus* conidia into BALB/c mice promotes mammary tumor growth through Treg amplification and systemic TIMP-1 elevation (Sohrabi et al, 2010). We used a C57BL/6 J mouse model of inhalation of *A. fumigatus* conidia. This

suggests that target cells of *A. fumigatus* may differ in different genetic backgrounds or tumor microenvironments.

In addition, we noted that *A. fumigatus* treatment leads to an increase in g-MDSCs in tumor tissue accompanied by a decrease in the number of CD8 T cells. The accelerated tumor progression as well as increased tumor angiogenesis caused by *A. fumigatus* was significantly suppressed after removal of g-MDSCs. However, the number of CD8 T cells was partially restored. These experimental results suggest that *A. fumigatus* affects tumor progression mainly through the pro-angiogenic function of MDSCs. However, there are limitations to our study. We note that the increased angiogenesis induced by *A. fumigatus* may lead to tumor dedifferentiation. To verify whether it is the increased angiogenesis that leads to dedifferentiation, we would also need to administer antivascular therapeutic drugs concurrently in the mouse model of *A. fumigatus* inhalation and then test the degree of differentiation of the tumors. Besides, different microbial strains have different virulence, attachment to endothelial cells, and pro-inflammatory activities. These differences may make various cells in the tumor area respond differently to microbes and turn on different cell signaling pathways. In future research, we will compare how different microbial strains affect blood vessel growth.

Moreover, mRNA-sequencing experiments revealed a significant upregulation of the *Slc7a11* in MDSCs following exposure to *A. fumigatus*. SLC7A11 is overexpressed in several cancer types, including glioma and non-small cell lung cancer, and serves as an independent prognostic factor (Koppula et al, 2021; Sun et al, 2022; Zhang et al, 2021). In primary brain tumors, ATF4 activation promotes angiogenesis and neuronal cell death and leads to iron death in an xCT-dependent manner (Kang et al, 2023), which is consistent with the results in this chapter. Inhibition of SLC7A11 reduced the pro-angiogenic effects of MDSCs induced by *A. fumigatus* in vivo. Mass spectrometry sequencing experiments showed that HMGB1, a protein associated with pro-angiogenesis (Li et al, 2023; Ohmori et al, 2011), interacted with SLC7A11. Inhibition of HMGB1 expression attenuated the pro-angiogenic effects induced by *A. fumigatus*.

Therefore, our results provide evidence that *A. fumigatus* can promote angiogenesis and accelerate disease progression by modulating MDSCs in the tumor microenvironment, rather than acting on endothelial or tumor cells. Mechanistically, SLC7A11 on MDSCs exerts pro-angiogenic effects through an interaction with HMGB1. Targeting SLC7A11 on MDSCs could be a promising therapeutic strategy for *A. fumigatus*-associated lung cancer.

    

# Methods

### Reagents and tools table

| Reagent/Resource | Reference or Source | Identifier or Catalog Number |
|---|---|---|
| **Experimental models** | | |
| A549 cells (*H. sapiens*) | Cell Bank at the China Academy of Science | SCSP-503 |
| SPCA1 cells (*H. sapiens*) | Cell Bank at the China Academy of Science | (Zhao et al, 2011) |
| HUVEC cells (*H. sapiens*) | Cell Bank at the China Academy of Science | (Chen et al, 2015) |
| 293 T cells (*H. sapiens*) | Cell Bank at the China Academy of Science | (Xu et al, 2022) |
| Lewis lung carcinoma (LLC) cell lines (*M. musculus*) | Cell Bank at the China Academy of Science | (Liu et al, 2016) |
| C57BL/6 J (*M. musculus*) | Model Animal Research Center of Nanjing University | N/A |
| **Recombinant DNA** | | |
| **Antibodies** | | |
| Ki67 | Immunoway | YM8189 |
| CD31 | Immunoway | YM8207 |
| VEGF-A | Abcam | ab52917 |
| CD34 | Immunoway | YM8525 |
| CD11b | Immunoway | YM8440 |
| panCK | Proteintech | 26411-1-AP |
| PE/Cy7 anti-mouse CD45 | BioLegend | 103113 |
| FITC anti-mouse CD11b | BioLegend | 101205 |
| APC anti-mouse Ly-6G/Ly-6C (Gr-1) | BioLegend | 108411 |
| APC anti-mouse Ly-6G | BioLegend | 127613 |
| PE anti-mouse Ly-6C | BioLegend | 128007 |
| APC-conjugated anti-mouse F4/80 | BioLegend | 123115 |
| FITC-conjugated anti-mouse CD11c | BioLegend | 117305 |
| APC anti-mouse MHC II | BioLegend | 116417 |
| APC anti-mouse CD3 | BioLegend | 100235 |
| FITC-conjugated anti-mouse CD4 | BioLegend | 130308 |
| FITC-conjugated anti-mouse CD8 | BioLegend | 100705 |
| APC anti-mouse CD25 | BioLegend | 113708 |
| PE-conjugated anti-mouse Foxp3 | BioLegend | 126403 |
| ASCT2 | Cell Signaling Technology | #5345 |
| SLC7A11 | Cell Signaling Technology | #98051 |
| β-actin | Cell Signaling Technology | #4970 |
| GLS | Proteintech Group | #12855 |
| GOT | Proteintech Group | #14886 |
| GCLC | Proteintech Group | #12601 |
| SLC7A11 | Abcam | ab175186 |

| Reagent/Resource | Reference or Source | Identifier or Catalog Number |
|---|---|---|
| HMGB1 | Genetex | GT348 |
| Flag-Tag | Proteintech Group | 66008-4-Ig |
| HA-Tag | Proteintech Group | 66006-2-Ig |
| Flag-Tag | Cell Signaling Technology | 14793S |
| HA-Tag | Cell Signaling Technology | 3724S |
| Goat Anti-Rabbit IgG H&L, Alexa Fluor® 647 | Invitrogen | A21235 |
| Goat Anti-Mouse IgG H&L, Alexa Fluor® 488 | Invitrogen | A11008 |
| Goat Anti-Rabbit IgG H&L, Alexa Fluor® 488 | Abclonal | AS073 |
| Goat Anti-Rabbit IgG (H + L) (AbFluor 555) | Immunoway | RS3411 |
| anti-mouse VEGF-A | Bioxcell | BE0399 |
| rat IgG2a isotype control | Bioxcell | BE0089 |
| Ultra-LEAF™ Purified anti-mouse Ly-6G | Biolegend | 127649 |
| Ultra-LEAF™ Purified Rat IgG2a, κ Isotype Ctrl | Biolegend | 400565 |
| InVivoMAb anti-rat Kappa Immunoglobulin Light Chain | Bioxcell | BE0122 |
| **Oligonucleotides and other sequence-based reagents** | | |
| Human GAPDH forward primer,5′-GTCTCCTCTGACTTCAACAGCG-3′ | This study | N/A |
| Human GAPDH reverse primer,5′-ACCACCCTGTTGCTGTAGCCAA-3′ | This study | N/A |
| Human EGF forward primer,5′-TGGATGTGCTTGATAAGCGG-3′ | This study | N/A |
| Human EGF reverse primer,5′-ACCATGTCCTTTCCAGTGTGT-3′ | This study | N/A |
| Human ANG-2 forward primer,5′-AACTTTCGGAAGAGCATGGAC-3′ | This study | N/A |
| Human ANG-2 reverse primer,5′-CGAGTCATCGTATTCGAGCGG-3′ | This study | N/A |
| Human VEGF-A forward primer,5′-TTGCCTTGCTGCTCTACCTCCA-3′ | This study | N/A |
| Human VEGF-A reverse primer,5′-GATGGCAGTAGCTGCGCTGATA-3′ | This study | N/A |
| Mouse β-actin forward primer,5′-TACCACCATGTACCCAGGCA-3′ | This study | N/A |
| Mouse β-actin reverse primer,5′-CTCAGGAGGAGCAATGATCTTGAT-3′ | This study | N/A |
| Mouse *Egf* forward primer,5′-AGAGCATCTCTCGGATTGACC-3′ | This study | N/A |
| Mouse *Egf* reverse primer,5′-CCCGTTAAGGAAAACTCTTAGCA-3′ | This study | N/A |
| Mouse *Ang2* forward primer,5′-AGAATAAGCAAGTCTCGCTTCC-3′ | This study | N/A |

| Reagent/Resource | Reference or Source | Identifier or Catalog Number |
|---|---|---|
| Mouse *Ang2* reverse primer,5'-TGAACCCTTTAGAGGCTCGGT-3' | This study | N/A |
| Mouse *Tgf-β* forward primer,5'-CTCCCGTGGCTTCTAGTGC-3' | This study | N/A |
| Mouse *Tgf-β* reverse primer,5'-GCCTTAGTTTGGACAGGATCTG-3' | This study | N/A |
| Mouse *Vegf-a* forward primer,5'-GCACATAGAGAGAATGAGCTTCC-3' | This study | N/A |
| Mouse *Vegf-a* reverse primer,5'-CTCCGCTCTGAACAAGGCT-3' | This study | N/A |
| *Aspergillus fumigatus* FISH probe sequence,5'cy3-TGACGGCCCGTTCCAG | This study | N/A |
| **Chemicals, Enzymes and other reagents** | | |
| Dulbecco's modified Eagle medium | Gibco | 11965092 |
| RPMI Medium 1640 | Gibco | 1875500BT |
| Opti-MEM™ | Gibco | 31985070 |
| Sulfasalazine | MCE | HY-14655 |
| *Aspergillus fumigatus*(AF293) | Tsinghua University, Beijing, China | N/A |
| Phosphatase inhibitor | MCE | HY-K0023 |
| Protease inhibitor | MCE | HY-K0010 |
| Liquid Sabouraud medium | Solarbio | L8300 |
| 4% paraformaldehyde solution, | Servicebio | G1101 |
| DAPI | SouthernBiotech | 0100-20 |
| Glycyrrhizin | MCE | HY-N0184 |
| BSA | Biofroxx | 9048-46-8 |
| Dewaxing solution | Biosharp | BL170A |
| Sodium citrate antigen repair solution | MXB | MVS-0101 |
| Hybridization solution | Servicebio | G3046 |
| 20xSSC | Servicebio | G3015 |
| Human VEGF-A ELISA Kit | DAKEWE | 121734096 |
| Mouse VEGF-A ELISA Kit | DAKEWE | 1217343 |
| RIPA Lysis Buffer | Solarbio | R0010 |
| True-Nuclear™ Transcription Factor BufferSet | BioLegend | 424401 |
| eBioscience™ Foxp3/Transcription Factor Staining Buffer Set | eBioscience | 00-5523-00 |
| TRIzol | Invitrogen | 15596018CN |
| Taq Pro Universal SYBR qPCRMaster Mix | Vazyme | Q712-02 |
| HisyGo RT Red SuperMix for qPCR (+gDNA Wiper) | Vazyme | RT101-01 |
| Anti-Ly-6G MicroBeads UltraPure | Miltenyi Biotec | 130-120-337 |
| CCK-8 Kit | Dojindo | CK04 |
| Crystal violet | Biosharp | BL802A |
| Capturem™ IP & Co-IP Kit | Takara | 635721 |

| Reagent/Resource | Reference or Source | Identifier or Catalog Number |
|---|---|---|
| IL-6 | Miltenyi Biotec | 130-096-683 |
| GM-CSF | Miltenyi Biotec | 130-095-746 |
| Matrigel | BD | 356234 |
| Lipofectamine RNAiMAX Transfection Reagent | Invitrogen | 13778150 |
| Lipofectamine 3000 | Invitrogen | L3000015 |
| Pierce Classic Magnetic Bead Immunoprecipitation/Co-Immunoprecipitation Kit | Thermo Scientific | 88804 |
| **Software** | | |
| Olympus Fluoviewver.3.0 software | OLYMPUS | N/A |
| FlowJo | https://www.flowjo.com/solutions/flowjo | N/A |
| PyMOL 2.3.2 | http://www.pymol.org/ | N/A |
| Prism software program (version 8.0; GraphPad Software) | https://www.graphpad.com/features | N/A |
| ImageJ | https://imagej.net/soft.ware/imagej/ | N/A |
| QuantStudio™ Real-Time PCR Software | https://www.thermofisher.cn/cn/zh/home/life-science/pcr/real-time-pcr/real-time-pcr-instruments/viia-7-real-time-pcr-system.html | N/A |
| BioRender | https://app.biorender.com/ | N/A |
| **Other** | | |
| Nebulizer | Yuwell | N/A |
| Olympus FV3000 confocal microscope | OLYMPUS | N/A |
| FACS Calibur flow cytometer | Becton Dickinson | N/A |
| ABI Vii 7 detection system | Thermo Fisher Scientific | N/A |
| CytoFlex | Beckman | N/A |
| Olympus VS200 slideview | OLYMPUS | N/A |

## Cell lines and cell culture

The human lung adenocarcinoma lines A549 and SPCA1, human umbilical vein endothelial cell line HUVEC, human embryonic kidney cell line 293T and Lewis lung carcinoma (LLC) cell line were obtained from the Cell Bank at the China Academy of Science. All cell lines were characterized using mycoplasma detection, DNA fingerprinting, isozyme detection, and cell viability analysis. No further authentication was conducted. Cells were cultured at 37 °C

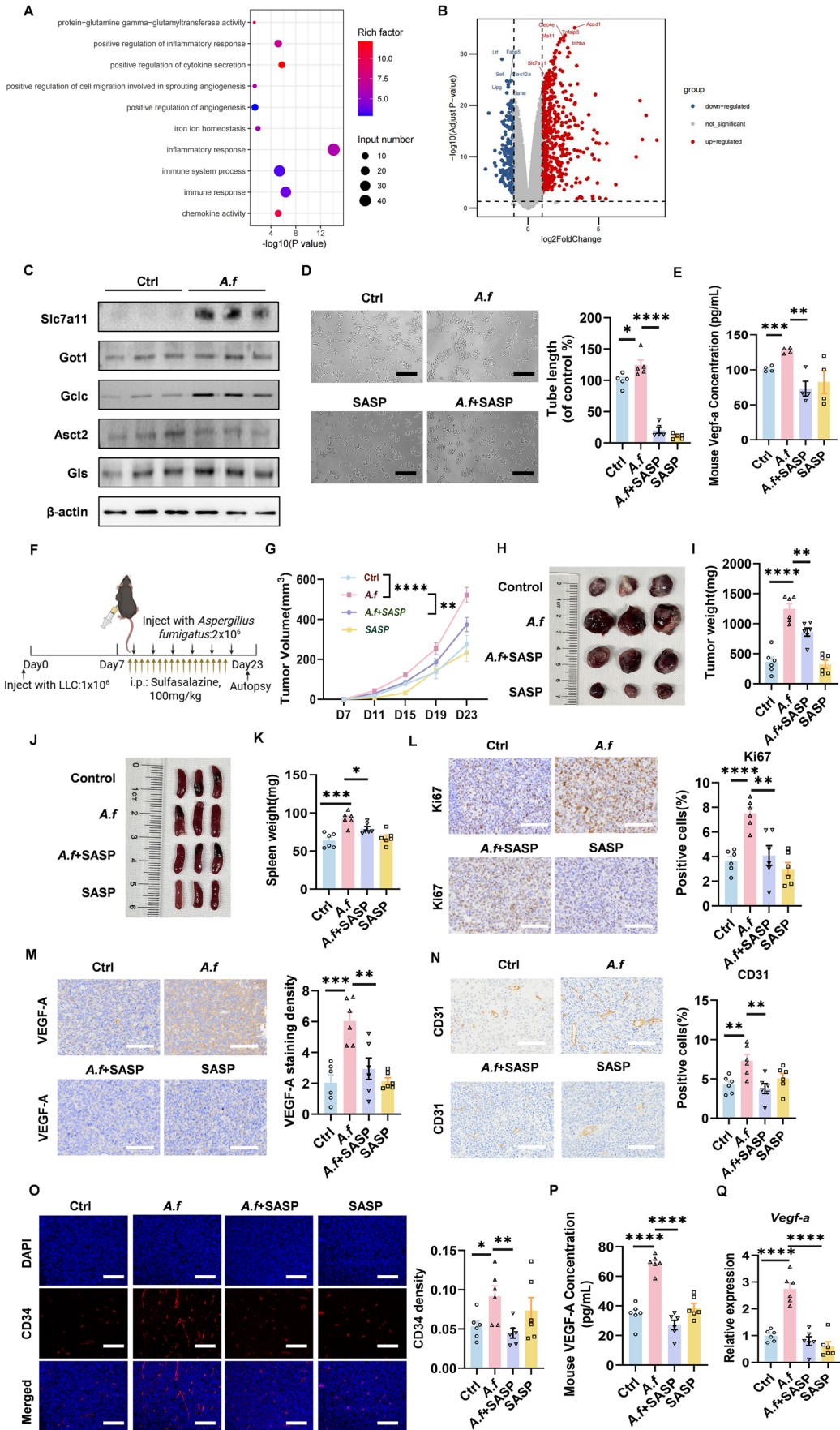

**Figure 5. SLC7A11 mediates the pro-angiogenic function of MDSCs induced by A. fumigatus.**

(A) MDSCs were treated with heat-inactivated *A.fumigatus* (MOI = 2) for 12 h compared with the control. GO enrichment map of up-regulated genes. (B) The volcano plot showed the differentially expressed genes of MDSCs. Red represents up-regulated genes and blue represents down-regulated genes ($n = 3$ biological replicates). (C) The proteins related to glutamine metabolism were detected by Western Blot. The inhibitor of Slc7a11, sulfasalazine (SASP) (2 μM), was added to the supernatant of the culture medium of MDSCs. Medium supernatants from MDSCs were collected and used as conditioned medium for HUVEC cells. (D) The lumen-forming capacity of HUVEC cells were assessed (scale bars, 200 μm) (Ctrl and *A. f*: $p = 0.0432$; *A. f* and *A. f* + SASP: $p < 0.0001$) ($n = 5$ biological replicates). (E) VEGF-A secretion in the culture supernatant was detected by ELISA (Ctrl and *A. f*: $p = 0.0001$; *A. f* and *A. f* + SASP: $p = 0.0024$) ($n = 4$ biological replicates). (F) SASP (100 mg/kg) was injected intraperitoneally daily in the LLC mouse model (Created with BioRender.com.). (G) The volume of tumor was assessed. (Ctrl and *A. f*: $p < 0.0001$; *A. f* and *A. f* + SASP: $p = 0.0046$) ($n = 6$ biological replicates). (H) Representative image of tumor isolates from different groups. (I) The mass of tumor was assessed (Ctrl and *A. f*: $p < 0.0001$; *A. f* and *A. f* + SASP: $p = 0.0060$) ($n = 6$ biological replicates). (J) Representative picture of spleens. (K) The mass of spleens was assessed (Ctrl and *A. f*: $p = 0.0003$; *A. f* and *A. f* + SASP: $p = 0.0181$) ($n = 6$ biological replicates). (L–N) Tumor tissues were stained for Ki67, VEGF-A, CD31 (scale bars, 100μm). (Ki67: Ctrl and *A. f*: $p < 0.0001$; *A. f* and *A. f* + SASP: $p = 0.0041$; VEGF-A: Ctrl and *A. f*: $p = 0.0004$; *A. f* and *A. f* + SASP: $p = 0.0065$; CD31: Ctrl and *A. f*: $p = 0.0071$; *A. f* and *A. f* + SASP: $p = 0.0057$) ($n = 6$ biological replicates). (O) Immunofluorescence images of CD34 in tumors (CD34: Ctrl and *A. f*: $p = 0.0264$; *A. f* and *A. f* + SASP: $p = 0.0087$) ($n = 6$ biological replicates). (P) VEGF-A in serum was assessed (Ctrl and *A. f*: $p < 0.0001$; *A. f* and *A. f* + SASP: $p < 0.0001$) ($n = 6$ biological replicates). (Q) *Vegf-a* gene expression in tumor tissues was measured by qPCR (Ctrl and *A. f*: $p < 0.0001$; *A. f* and *A. f* + SASP: $p < 0.0001$) ($n = 6$ biological replicates). Data information: Data with error bars are represented as mean ± SEM. Normality was assessed using the Shapiro-Wilk test. For normally distributed data, two-group comparisons were performed using the unpaired t-test or two-way ANOVA. For non-normal data, the Mann-Whitney U test was used. Tumor growth curves were analyzed by two-way ANOVA. $*p < 0.05$, $**p < 0.01$, $***p < 0.001$, $****p < 0.0001$ as determined by unpaired Student's t-test or two-way ANOVA. Source data are available online for this figure.

in a humidified atmosphere containing 5% CO2. Specifically, A549, 293T, and LLC cells were cultured in Dulbecco's modified Eagle medium (DMEM; Gibco). SPCA1 and HUVEC cells were cultured in Roswell Park Memorial Institute (RPMI)-1640 medium (Gibco). All medium were supplemented with 10% fetal bovine serum (FBS; Gibco), 1% penicillin, and 1% streptomycin (Gibco).

## Animals and mouse model

To establish the Lewis lung cancer mouse model, six-week-old male C57BL/6J mice were purchased from the Model Animal Research Center of Nanjing University and bred in independent ventilation system under specific pathogen-free conditions.

$1 \times 10^6$ LLC cells were administrated by subcutaneous injection into the right axilla of each mouse. Mice were randomly divided into different groups a week after injection ($n = 6$ for each group). Peritumoral injection of *A. fumigatus* ($2 \times 10^6$) three times per week for two weeks.

An orthotopic lung cancer model was established in C57BL/6J mice. 50 μl of cell suspension containing $5 \times 10^5$ LLC cells mixed with Matrigel (BD, 356324) was injected into the left lung of mice. All mice were euthanized on day 21 after tumor cell injection.

Resuspend *A. fumigatus* spores in sterile PBS to a concentration of $1 \times 10^8$ spores/ml. Connect a nebulizer (Yuwell, 403 M) to a closed container and allow the mice to inhale the spores for 75 s. After an interval of 1 h, the mice were allowed to inhale the spores for another 75 s. Inhalation experiments were performed every two days.

In the first week, anti-Ly6G (1A8, Biolegend) was injected intraperitoneally with 25 μg/mouse. In the second week, anti-Ly6G (1A8, Biolegend) was injected intraperitoneally with 50 μg/mouse. Anti-rat Kappa immunoglobulin (MAR 18.5, Bioxcell) was injected intraperitoneally every two days, 50 μg/mouse. The corresponding isotype control (Biolegend) was injected intraperitoneally with 50 μg/mouse. When mice are injected sequentially with both antibodies, an interval of 2 h is required. Sulfasalazine was intraperitoneal injection at a dose of 100 mg/kg daily. The tumor volume was measured every 3 days until the endpoint and calculated according to the equation volume = length × width$^2$ ×

1/2. After three weeks, all mice were euthanized. Spleens and tumors were collected and weighed. All animal operations were performed in compliance with the "National Institutes of Health Guidelines for the Care and Use of Laboratory Animals" and were approved by the Institutional Animal Care and Use Committee of Nanjing University Medical School (SYXK2024-0067).

## Fungus strain and growth conditions

*Aspergillus fumigatus* (AF293) was kindly provided by Dr. Xin Lin (Tsinghua University, Beijing, China). *A. fumigatus* was cultured in liquid Sabouraud medium at 28 °C at 200 rpm for 24 h and then at 37 °C at 150 rpm. Heat inactivation was performed in a 95 °C metal bath for 45 min.

## Histology and immunohistochemical (IHC) analysis

The tumor tissues were fixed in a 4% paraformaldehyde solution, and embedded in paraffin. Hematoxylin-eosin staining was performed on paraffin-embedded tumor tissue sections. Histological scoring evaluated tumor cell morphology, tumor necrosis, infiltration of inflammatory cells, and invasion. The scoring method is detailed in the histological scoring criteria in Appendix Table S3. The indicator Ki67 is used to detect the proliferative activity of tumor cells. The indicator CD31 is used to detect the presence of endothelial cells and is used to assess tumor angiogenesis. For IHC staining, the indicated antibodies (Ki67, Immunoway, 1:200; CD31, Immunoway, 1:400; VEGF-A, Abcam, 1:100) were used to stain tumor sections. Stained sections were examined under a light microscope.

## Immunofluorescence

For immunofluorescence, frozen tumor sections (7 μm) were prepared. Anti-CD34 was used to stain endothelial cells (CD34, Immunoway, 1:100). Then the slides were incubated with Goat Anti-Rabbit IgG (H + L) (AbFluor 555) (Immunoway) secondary antibody at a 1:500 dilution for 1 h at room temperature in the dark. DAPI was used to stain nuclei (SouthernBiotech). A confocal

laser scanning microscope was used for detection. To detect the co-localization of SLC7A11 and HMGB1 in MDSCs, we used polylysine-coated coverslips, which MDSC cells can adhere to. The HMGB1 inhibitor Glycyrrhizin (1 μM) (MCE, #HY-N0184) was added to the culture medium supernatant of MDSCs. Cells were fixed in 4% paraformaldehyde and incubated in a blocking buffer (5% BSA in PBS) for 1 h at room temperature. Cells were then incubated with primary antibodies (HMGB1, Genetex, 1:500) (SLC7A11, Abcam, 1:500) overnight at 4 °C. After rinsing thrice with PBS, cells were incubated with secondary antibodies (Goat Anti-Rabbit IgG H&L, Alexa Fluor® 647, Invitrogen, 1:200) (Goat Anti-Mouse IgG H&L, Alexa Fluor® 488, Invitrogen, 1:200). Finally, nuclei were stained with DAPI (SouthernBiotech). We took pictures with an Olympus FV3000 confocal microscope (Nanjing, China). Images were analyzed using Olympus Fluoviewver.3.0 software.

## FISH

Tissues were collected and rinsed in PBS and immediately fixed with in situ hybridization fixative for more than 12 h. After completion of tissue fixation, the tissues were dehydrated and immersed in wax for embedding, and sliced by paraffin slicer. 80 °C oven baked slices overnight. The paraffin sections were dewaxed with dewaxing solution as well as alcohol and soaked in DEPC water. Tissue sections were antigenically repaired using sodium citrate antigen repair solution. Delivery Pre-hybridization solution was added dropwise and incubated at 40 °C for 1 h (Servicebio, G3015-100M). Probe-containing hybridization solution was added dropwise and hybridization was performed at 37 °C overnight. Wash with SSC solution after hybridization. The sections were blocked with BSA for 30 min. the primary antibody (CD11b, Immunoway, 1:500; panCK, Proteintech, 1:200; CD31, Immunoway, 1:500) was incubated at 4 °C overnight. After washing with PBS, use fluorescein-labeled secondary antibody and incubate at room temperature for 50 min (Goat Anti-Rabbit IgG H&L, Alexa Fluor® 488, Abclonal, 1:100). after washing with PBS, add DAPI (Bioworld) staining solution dropwise and seal the film with sealer. *Aspergillus fumigatus* FISH probe sequences are referenced from Volker Rickerts et al (Rickerts et al, 2011). *Aspergillus fumigatus* FISH probe sequence: 5'cy3-TGACGGCCCGTTCCAG.

## Detection of cytokines

The blood samples were centrifuged at 3000 rpm for 20 min to separate the serum. All serum samples were analyzed within 24 h. The cell samples were centrifuged at $300 \times g$ for 10 min to separate the supernatant. The concentration of VEGF-A was evaluated with the murine or human ELISA kit (DAKAWE, China) according to manufacturer's instructions.

## Flow cytometric analysis

Prepare a single-cell suspension of tissue (Qu et al, 2019). Tumor, spleen, and bone marrow tissues were isolated and ground using the bottom of a syringe. Cells were collected and centrifuged. The supernatant was discarded, and the cells were lysed with red blood cell lysis buffer and washed with PBS. Take $2–3 \times 10^6$ cells and dispense them into flow-through tubes. To determine the number of MDSCs, the single-cell suspensions were stained with fluorescently conjugated antibodies: PE/Cy7 anti-mouse CD45 (BioLegend, 2.5 μg/ml), FITC anti-mouse CD11b (BioLegend, 2.5 μg/ml), APC anti-mouse Ly-6G/Ly-6C (Gr-1) (BioLegend, 2.5 μg/ml), APC anti-mouse Ly-6G (BioLegend, 5 μg/ml) and PE anti-mouse Ly-6C (BioLegend, 5 μg/ml). For the detection of macrophages, the single-cell suspensions were stained with fluorescent conjugated antibodies: PE/Cy7 anti-mouse CD45 (BioLegend, 2.5 μg/ml), FITC anti-mouse CD11b (BioLegend, 2.5 μg/ml) and APC-conjugated anti-mouse F4/80 (BioLegend, 5 μg/ml). For the detection of DCs, the single-cell suspensions were stained with fluorescently conjugated antibodies: PE/Cy7 anti-mouse CD45 (BioLegend, 2.5 μg/ml), FITC-conjugated anti-mouse CD11c (BioLegend, 2.5 μg/ml) and APC anti-mouse MHC II (BioLegend, 2.5 μg/ml). For the detection of CD4+T cells, the single-cell suspensions were stained with fluorescently conjugated antibodies: PE/Cy7 anti-mouse CD45 (BioLegend, 2.5 μg/ml), APC anti-mouse CD3 (BioLegend, 2.5 μg/ml) and FITC-conjugated anti-mouse CD4 (BioLegend, 1.25 μg/ml). For the detection of CD8+T cells, the single-cell suspensions were stained with fluorescently conjugated antibodies: PE/Cy7 anti-mouse CD45 (BioLegend, 2.5 μg/ml), APC anti-mouse CD3 (BioLegend, 2.5 μg/ml) and FITC-conjugated anti-mouse CD8 (BioLegend, 1.25 μg/ml). For the detection of Treg cells, the single-cell suspensions were stained with fluorescently conjugated antibodies: PE/Cy7 anti-mouse CD45 (BioLegend, 2.5 μg/ml), APC anti-mouse CD25 (BioLegend, 2.5 μg/ml), FITC-conjugated anti-mouse CD4 (BioLegend, 1.25 μg/ml) and PE-conjugated anti-mouse Foxp3 (BioLegend, 2.5 μg/ml). Flow cytometry of tumor-derived cells: Add the corresponding flow antibody to each flow tube and incubate for 15 min at room temperature. Wash cells with 1 ml PBS, centrifuge at 300 g for 5 min and discard the supernatant. Add 200 μl PBS for resuspension and flow cytometry for detection. Intracellular Ly6G was assayed with reference to the method of Gael Boivin et al (Boivin et al, 2020). Briefly, after completion of Ly6G staining on the cell surface, cells were fixed and permeabilized (eBioscience). Then, the cells were incubated with anti-Ly6G for 15 min at 4 °C. Intracellular staining was diluted twice less than membrane staining. True-Nuclear™ Transcription Factor BufferSet (BioLegend) was used to detect Treg cells. Alternatively, add 200 μl of paraformaldehyde solution at a concentration of 1%, store at 4 °C and assay within one week. Cells were detected using a FACS Calibur flow cytometer (Becton Dickinson, Franklin Lakes, NJ). Data were analyzed using FlowJo software (Treestar, Inc., San Carlos, CA).

## RT-PCR and Q-PCR analysis

Total RNAs of tumor tissue or cells were extracted with TRIzol Reagent (Invitrogen, Carlsbad, CA) and were reverse-transcribed into cDNA using oligo (dT) primer. StepOne Plus or an ABI Vii 7 detection system (Applied Biosystems, Thermo Fisher Scientific, USA) with SYBR Green PCR master mix solution was used for qPCR assays. The relative expression of the gene was calculated with the formula of $2(-\Delta CT)$. β-actin was used as the internal control. The oligonucleotide primers used for Quantitative real-time PCR amplification are listed in Reagents and Tools Table.

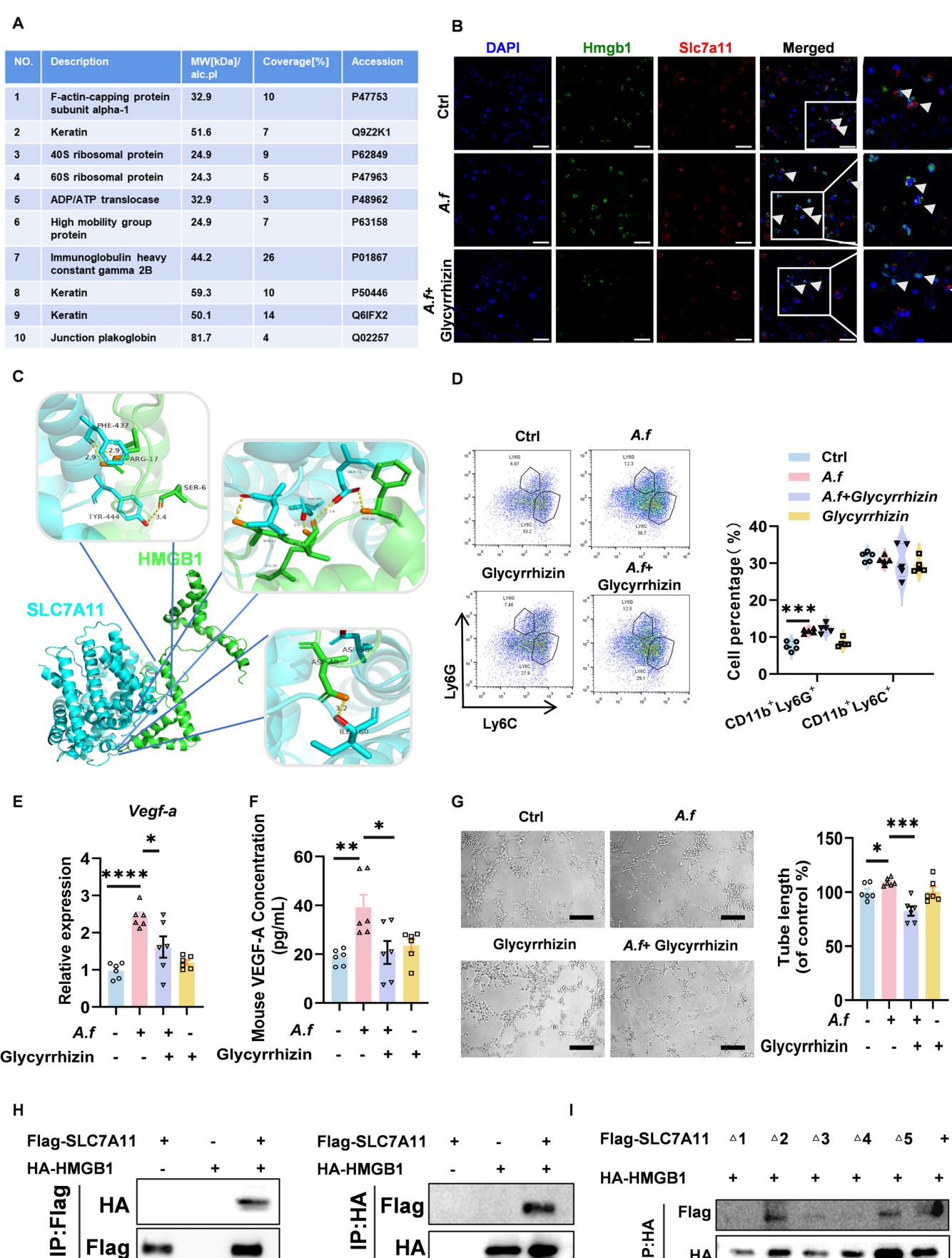

**Figure 6. HMGB1 interacts with SLC7A11 in MDSCs.**

(A) MDSCs treated with heat-inactivated *A. fumigatus* (MOI = 2) for 24 h. COIP/MS analysis of SLC7A11-binding proteins, part of the interaction proteins was shown. (B) Colocalization of HMGB1 and SLC7A11 was assessed by confocal microscopy (scale bars, 20 µm). (C) The structure of SLC7A11 bound to HMGB1 was simulated by the online docking tool ZDOCK, and the binding results were visualized by Pymol. The amino acid Phe437 in SLC7A11 forms two hydrogen bonds with the amino acid Arg17 in HMGB1, and the interaction distance is 2.9 Å. The amino acid Tyr444 in SLC7A11 forms one hydrogen bond with the amino acid Ser6 in HMGB1, and the interaction distance is 3.4 Å. The amino acid Gln71 in SLC7A11 forms one hydrogen bond with the amino acid Phe45 in HMGB1, and the interaction distance is 3 Å. The amino acid Gln71 in SLC7A11 forms one hydrogen bond with the amino acid Val43 in HMGB1, and the interaction distance is 3.4 Å. The amino acid Leu299 in SLC7A11 forms one hydrogen bond with the amino acid Ser42 in HMGB1, and the interaction distance is 2.5 Å. The amino acid Asp386 in SLC7A11 forms one hydrogen bond with the amino acid Lys35 in HMGB1, and the interaction distance is 2 Å. The amino acid Ile160 in SLC7A11 forms one hydrogen bond with the amino acid Asp40 in HMGB1, and the interaction distance is 3.2 Å. (D) The inhibitor of HMGB1, Glycyrrhizin (1 µM), was added to the supernatant of the culture medium of MDSCs. Proportions of total MDSCs, G-MDSCs and M-MDSCs were detected by flow cytometry (Ctrl and *A. f*: $p = 0.0002$) ($n = 5$ biological replicates). (E) The gene expression of *Vegf-a* in MDSCs was detected by qPCR (Ctrl and *A. f*: $p < 0.0001$; *A. f* and *A. f* + Glycyrrhizin: $p = 0.0265$) ($n = 6$ biological replicates). (F) VEGF-A secretion in the culture supernatant was detected by ELISA (Ctrl and *A. f*: $p = 0.0032$; *A. f* and *A. f* + Glycyrrhizin: $p = 0.0244$) ($n = 6$ biological replicates). (G) Medium supernatants from MDSCs were collected and used as a conditioned medium for HUVEC cells. The lumen-forming capacity of HUVEC cells were assessed (scale bars, 200 µm) (Ctrl and *A. f*: $p = 0.0204$; *A. f* and *A. f* + Glycyrrhizin: $p = 0.0001$) ($n = 6$ biological replicates). (H) 293T cells transfected with Flag-tagged SLC7A11 and HA-tagged HMGB1. After 24 h, cell lysate was used for IP with anti-Flag (left) or anti-HA (right). (I) 293T cells transfected with Flag-SLC7A11 WT or its mutant (mutant1: Asp368, mutant2: Phe437, mutant3: Leu299, mutant: Gln71, mutant: Tyr 444) and HA-HMGB1, followed by IP with HA. Data information: Data with error bars are represented as mean ± SEM. Normality was assessed using the Shapiro-Wilk test. For normally distributed data, two-group comparisons were performed using the unpaired t-test. For non-normal data, the Mann-Whitney U test was used. *$p < 0.05$, **$p < 0.01$, ***$p < 0.001$, ****$p < 0.0001$ as determined by unpaired Student's t-test or Mann-Whitney U test. Source data are available online for this figure.

## Generation of BM-derived MDSCs

Bone marrow-derived MDSCs were extracted from C57BL/6J mice from 6 to 8 weeks old. Myelocytes were isolated from mice by flushing femurs and tibiae. Then, Myelocyte was centrifuged and resuspended in RPMI 1640 medium supplemented with murine interleukin 6 (IL-6) and granulocyte-macrophage colony stimulating factor (GM-CSF), both 40 ng/mL (Miltenyi Biotec, Bergisch Gladbach, Germany). Cells were cultured for 4 days.

## MACS cell separation

The Anti-Ly-6G MicroBeads UltraPure (130-120-337) (Miltenyi Biotec) was applied to isolate Ly-6G$^+$ cells from mouse tumor tissue according to the instructions.

## Western Blot analysis

Protein concentration was measured after cell lysis. Following antibodies were used (all from Cell Signaling Technology): ASCT2 (#5345), SLC7A11 (#98051) and β-actin (#4970). Following antibodies were used (all from Proteintech Group): GLS (#12855), GOT (#14886), GCLC (#12601), Flag-Tag (66008-4-lg) and HA-Tag (66006-2-lg).

## Cell viability

Cells were seeded into 96-well plates in appropriate proportions, drug was added after cells were adapted and cultured for 24 h. The 96-well plate was centrifuged at $300 \times g$ for 5 min and the medium was discarded. Then, 100 µL (10% CCK-8)/well of culture medium was added with pipette discharge and cultured for 1–4 h. At 450 nm and 620 nm, the absorbance was measured.

## Wound healing assay

Wounded-monolayer cells were washed two or three times to remove detached cells. The initial size of the wound on the monolayer was determined using inverted microscopy immediately after the cells were washed. After 24 and 32 h of incubation in the cell supernatants stimulated with *A. fumigatus* wound closure was calculated as the percentage of the remaining initial wound area.

## Transwell migration and invasion assay

Cell migration and invasion ability was assessed using 12-well Transwell chambers with 8 µm pore size (Corning Incorporated, USA). For invasion experiments, Matrigel was added to the membrane side of the transwell. The Matrigel gel was diluted 1:6 with serum-free cell culture medium at 4 °C. 100 µl of diluted Matrigel gel was taken and added to the Transwell small chamber, and the gel was incubated for 3 h in the incubator at 37 °C. Add 600 µl of medium containing 10% FBS to the lower chamber. Cells were suspended in serum-free medium and the cell density was adjusted to $1 \times 10^5$ cells/ml. 200 µL of cell suspension was added to the upper chamber. After 24–48 h of culture, cells were fixed with 4% paraformaldehyde for 20 min, stained with crystal violet (Solarbio, Beijing, China) for 30 min and washed to prepare for imaging.

## Conditioned medium

MDSCs were stimulated with heat-inactivated *A. fumigatus* (MOI = 2) for 24 h. The supernatant was collected by centrifugation at $300 \times g$ for 10 min. To neutralize VEGF-A in the supernatant, the supernatant was pre-incubated with VEGF-A neutralizing antibody (2 µg/ml) (Bioxcell, USA) or the isotype control (2 µg/ml) (Bioxcell, USA) for 2 h at 37 °C to neutralize VEGF-A.

## Tube formation assay

The Matrigel was dissolved at 4 °C. On the ice surface, the Matrigel matrix glue was diluted with culture medium in a 1:1 ratio. Subsequently, 50 µl of the Matrigel-culture medium mixture was added to a 96-well plate, ensuring that the entire process was free of

contamination. The 96-well plates were incubated at 37 °C for 20–30 min until solidified. During the waiting period, the HUVEC cells were digested. Once the glue had solidified, 100 µl of the suspension containing $3 \times 10^4$ cells was added to each well of the 96-well plate. Samples were photographed at 6 h.

## Protein docking prediction

Using the ZDOCK 3.0.2 technology of the protein docking algorithm to make a prediction. The score function was used to rigidly dock the two proteins, and the docking produced the TOP 10 conformations. The top-1 conformation was used for graphical analysis. PyMOL 2.3.2 was used to display the three-dimensional graph of the protein-binding mode.

## UID mRNA sequencing

The TRIzol reagent (Invitrogen, Carlsbad, CA) was used to isolate total RNA from MDSCs ($1 \times 10^6$). Then, total RNAs were used for the library preparation of stranded RNA sequencing using the KCDigital™ Stranded mRNA Library Prep Kit for Illumina® (Wuhan Seqhealth Co., Ltd., China) based on the manufacturer's instruction.

## Small interfering RNA (siRNA) and plasmid transfection

MDSCs were plated in 12-well plates at appropriate density, 24 h before transfections. Cells were transfected with VEGF-A-targeting siRNA or negative control siRNA (NC) (RiboBio, China) using Lipofectamine RNAiMAX Transfection Reagent (Invitrogen, USA) according to the manufacturer's protocol. The siRNA and transfection reagent were diluted in Opti-MEM Medium before complex formation. After 48 h, cells were harvested for detection and following experiments.

At 70% confluency, 293T cells were transfected with plasmid DNA (Zebrafish Biotech, China) using Lipofectamine 3000 according to the manufacturer's protocol, with complexes formed in Opti-MEM medium. DNA-lipid complexes were added dropwise to cells in Opti-MEM medium. Following 4–6 h of incubation, the transfection mixture was replaced with fresh RPMI-1640 medium supplemented with 10% FBS. Cells were cultured for an additional 24 h prior to analysis or harvest. All incubations were performed at 37 °C under 5% $CO_2$.

## Immunoprecipitation, Co-IP and LC-MS

Immunoprecipitation experiments were used with Capturem™ IP & Co-IP Kit (Takara, #635721). To identify the interacting cellular protein targets of SLC7A11, we carried out Immunoprecipitation experiments using anti-SLC7A11 (Abcam, #ab175186) followed by LC-MS/MS. For LC-MS/MS protein identification, the SDS-PAGE gels were conducted coomassie brilliant blue staining, each lane was cut into gel slices and LC-MS/MS analysis at Shanghai Luming Biological Technology Co., Ltd (Shanghai, China).

Co-immunoprecipitation (Co-IP) experiments were performed using the Pierce Classic Magnetic Bead Immunoprecipitation/Co-Immunoprecipitation Kit (Thermo Scientific, #88804). HA-Tag(#3724S) and Flag-Tag(#14793S) antibodies were used from Cell Signaling Technology.

## Patient samples

This study included 15 patients with lung cancer who were treated at the Affiliated Hospital of Nanjing University Jinling Hospital. Lung cancer tissue and adjacent lung tissue were obtained during surgery. The clinical characteristics of the patients are summarized in Appendix Table S4. All studies involving human samples were approved by the Ethics Committee of the School of Medicine, Nanjing University (OAP20250106001), and all participants provided written informed consent.

## Statistical analysis

All experimental results were repeated more than three times. All quantitative data are averages of at least three biological replicates. Normality was assessed for each comparison group using the Shapiro-Wilk test. Based on this assessment, two-group comparisons used the unpaired t-test (normal) or Mann-Whitney U test (non-normal). For tumor growth curves, involving repeated measures, we employed two-way ANOVA analysis. Statistical analysis was performed using the Prism software program (version 8.0; GraphPad Software). $P < 0.05$ was defined as statistical significance. $*P < 0.05$, $**P < 0.01$, $***P < 0.001$, $****P < 0.0001$, and ns indicating no significance.

## Data availability

The raw UID mRNA sequence data generated in this study were deposited in the NCBI's Sequence Read Archive (SRA) under BioProject accession (PRJNA1279862). These data can be accessed through the link: https://www.ncbi.nlm.nih.gov/Traces/study/?acc=SRP593284&o=acc_s%3Aa. LC-MS data tables are provided in Appendix Table S1 and Appendix Table S2.

The source data of this paper are collected in the following database record: biostudies:S-SCDT-10_1038-S44319-025-00627-x.

## Peer review information

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

## Acknowledgements

This work is supported by grants from the Natural Science Foundation of Jiangsu Province (BK20211508), National Natural Science Foundation of China (82273011), the Fundamental Research Funds for the Central Universities (021414380472).

## Author contributions

**Wei Qu**: Conceptualization; Software; Formal analysis; Supervision; Validation; Investigation; Visualization; Writing—original draft; Writing—review and editing. **Zelin Wang**: Data curation; Validation; Investigation. **Tianchen Zhu**: Investigation; Writing—review and editing. **Huiyue Cui**: Methodology. **Ziqian Bing**: Data curation; Formal analysis; Validation; Investigation; Methodology. **Sunan Shen**: Conceptualization. **Yi Shen**: Resources; Methodology. **Shaorong Yu**: Resources; Methodology. **Hongqin Zhuang**: Conceptualization; Formal analysis; Methodology. **Tingting Wang**: Conceptualization; Resources; Data curation; Supervision; Funding acquisition; Methodology; Project administration.

Source data underlying figure panels in this paper may have individual authorship assigned. Where available, figure panel/source data authorship is listed in the following database record: biostudies:S-SCDT-10_1038-S44319-025-00627-x.

## Disclosure and competing interests statement

The authors declare no competing interests.

# Expanded View Figures

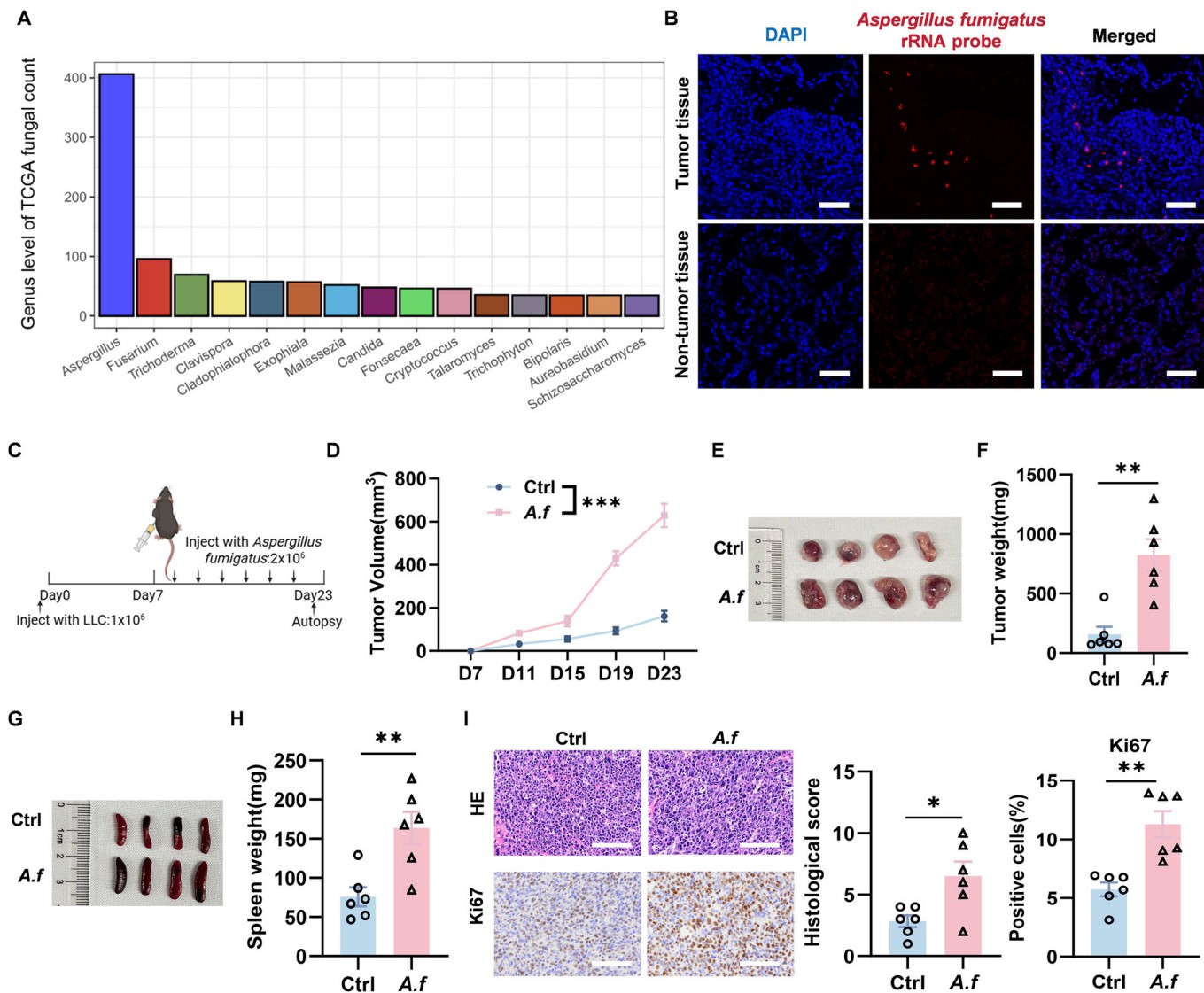

**Figure EV1.** *A. fumigatus* **promotes the development of lung cancer.**

(A) The distribution of different fungal genera in LUAD was analyzed based on the decontaminated TCGA fungal count data. (B) Detection of *A. fumigatus* in human lung cancer by FISH (scale bars, 50 µm). (C) Lewis lung cancer cells ($1 \times 10^6$) were injected subcutaneously into the right flank of mice ($n = 6$ per group). Live *A. fumigatus* ($2 \times 10^6$) was injected peritumorally three times per week for two weeks. All tumor-bearing mice were executed on day 23 (Created with BioRender.com). (D) Tumor-volume curve was calculated after injection every four days ($p = 0.0006$) ($n = 6$ biological replicates). (E) Representative images of tumors from different groups. (F) Tumors isolated from mice were weighed ($p = 0.0011$) ($n = 6$ biological replicates). (G) Representative images of spleens from different groups. (H) Spleens isolated from mice were weighed ($p = 0.0046$) ($n = 6$ biological replicates). (I) Histological analysis of tumors was shown by HE staining (scale bars, 100 µm) ($p = 0.0161$) Immunohistochemical analysis of Ki67 in tumors (scale bars, 100 µm) ($p = 0.0014$) ($n = 6$ biological replicates). Data information: Data with error bars are represented as mean ± SEM. Tumor growth curves were analyzed by two-way ANOVA. Other data were analyzed using unpaired Student's t-test. $*p < 0.05$, $**p < 0.01$, $***p < 0.001$ as determined by unpaired Student's t-test or two-way ANOVA.

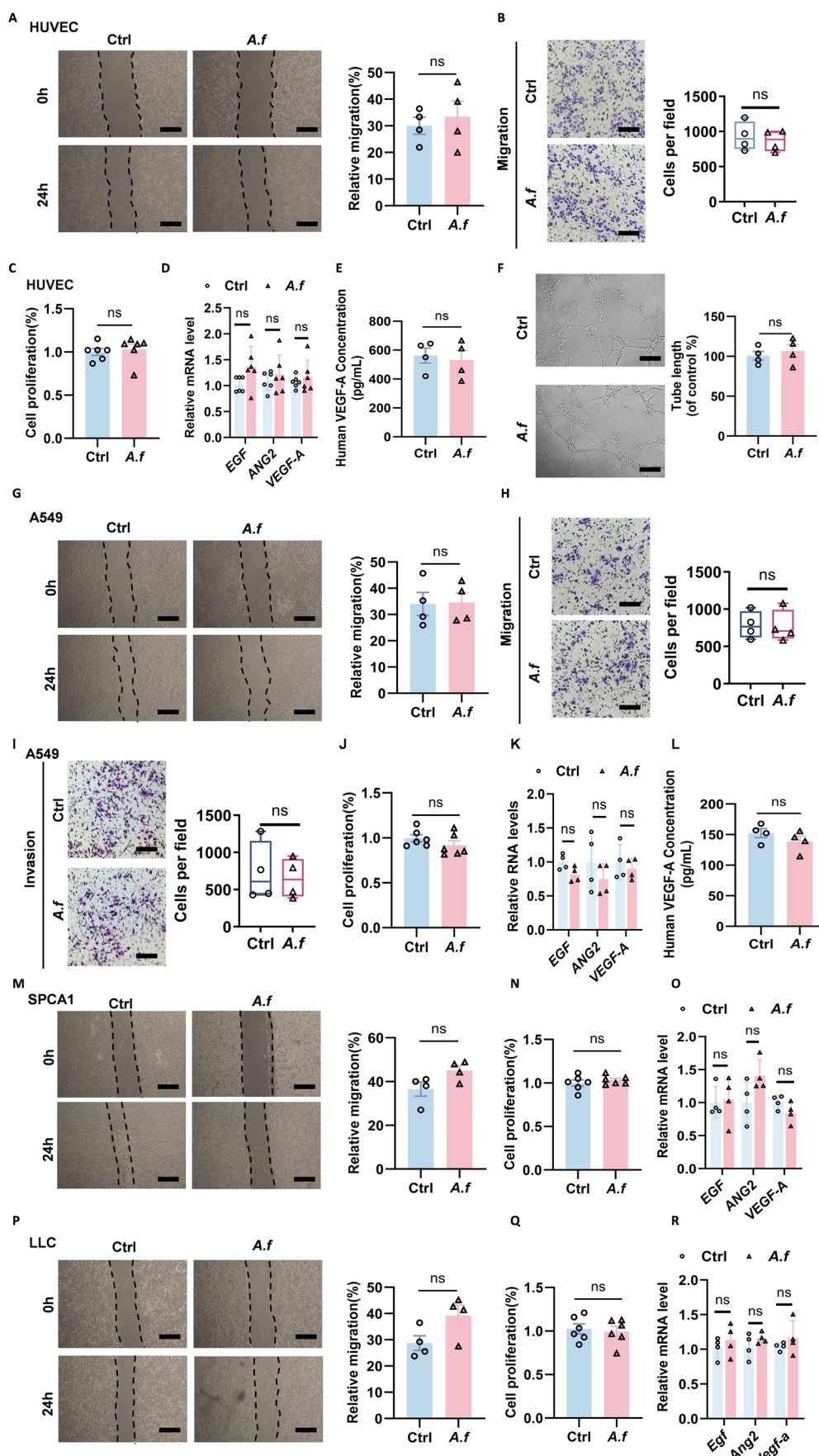

◀ **Figure EV2. *A. fumigatus* has no effect on migratory, cell viability and pro-angiogenic function of endothelial cells and lung cancer cells.**

Heat-inactivated *A. fumigatus* (MOI = 2) was used in the co-culture with HUVEC cells. (**A, B**) The migratory ability of HUVEC cells was examined by scratch assay (scale bars, 500 μm) ($p = 0.6314$) and Transwell migration assay (scale bars, 200 μm) ($p = 0.6492$). For Ctrl in box plot: Minima = 728; Maxima = 1197; Centre = 895; Box bounds = (774.5, 1083); Lower Whisker = 728; Upper Whisker = 1197; (25th percentile, 75th percentile) = (774.5, 1083). For *A. f* in box plot: Minima = 700; Maxima = 1001; Centre = 885.5; Box bounds = (741, 995); Lower Whisker = 700; Upper Whisker = 1001 (25th percentile, 75th percentile) = (741, 995) ($n = 4$ biological replicates). (**C**) The cell viability of HUVEC cells was detected using CCK8 ($p = 0.3095$) ($n = 6$ biological replicates). (**D**) The gene expression related to angiogenesis was detected by qPCR. (*EGF*: $p = 0.0649$; *ANG2*: $p = 0.4822$; *VEGF-A*: $p = 0.4939$) ($n = 6$ biological replicates). (**E**) ELISA analysis of VEGF-A secretion in the culture supernatant ($p = 0.7275$) ($n = 4$ biological replicates). (**F**) The lumen-forming capacity of HUVEC cells was examined by tube-formation assay (scale bars, 200 μm) ($p = 0.5628$) ($n = 4$ biological replicates). Heat-inactivated *A. fumigatus* (MOI = 2) was used in the co-culture with A549 cells. (**G, H**) The migratory ability was examined by scratch assay (scale bars, 500 μm) ($p = 0.9337$) and Transwell migration assay (scale bars, 200 μm) ($p = 0.9053$). For Ctrl in box plot: Minima = 594; Maxima = 1019; Centre = 764; Box bounds = (649, 921.5); Lower Whisker = 594; Upper Whisker = 1019; (25th percentile, 75th percentile) = (649, 921.5). For *A. f* in box plot: Minima = 583; Maxima = 1078; Centre = 705; Box bounds = (632.5, 903); Lower Whisker = 583; Upper Whisker = 1078; (25th percentile, 75th percentile) = (632.5, 903) ($n = 4$ biological replicates). (**I**) The invasion ability was examined by Transwell invasion assay (scale bars, 200 μm) ($p = 0.7494$). For Ctrl in box plot: Minima = 428; Maxima = 1284; Centre = 608; Box bounds = (438, 1026); Lower Whisker = 428; Upper Whisker = 1284; (25th percentile, 75th percentile) = (438, 1026). For *A. f* in box plot: Minima = 387; Maxima = 951; Centre = 634.5; Box bounds = (430.5, 873); Lower Whisker = 387; Upper Whisker = 951; (25th percentile, 75th percentile) = (430.5, 873) ($n = 4$ biological replicates). (**J**) The cell viability was assessed by CCK8 ($p = 0.2511$) ($n = 4$ biological replicates). (**K**) The gene expression related to angiogenesis was detected by qPCR (*EGF*: $p = 0.0565$; *ANG2*: $p = 0.3317$; *VEGF-A*: $p = 0.5663$) ($n = 4$ biological replicates). (**L**) VEGF-A secretion was analyzed by ELISA ($p = 0.2824$) ($n = 4$ biological replicates). Heat-inactivated *A. fumigatus* (MOI = 2) was used in the co-culture with SPCA1 cells. (**M**) The migratory ability was examined by scratch assay (scale bars, 500 μm) ($p = 0.0735$) ($n = 4$ biological replicates). (**N**) The cell viability was assessed by CCK8 ($p = 0.4244$) ($n = 6$ biological replicates). (**O**) The gene expression related to angiogenesis was detected by qPCR. (*EGF*: $p = 0.6857$; *ANG2*: $p = 0.0862$; *VEGF-A*: $p = 0.1387$) ($n = 4$ biological replicates). Heat-inactivated *A. fumigatus* (MOI = 2) was used in the co-culture with LLC cells. (**P**) The migratory ability was examined by scratch test (scale bars, 500 μm) ($p = 0.0747$) ($n = 4$ biological replicates). (**Q**) The cell viability was assessed by CCK8 ($p = 0.7188$) ($n = 6$ biological replicates). (**R**) The gene expression related to angiogenesis was detected by qPCR. (*Egf*: $p = 0.4545$; *Ang2*: $p = 0.2894$; *Vegf-a*: $p = 0.3724$) ($n = 4$ biological replicates). Data information: Data with error bars are represented as mean ± SEM. Normality was assessed using the Shapiro-Wilk test. Two-group comparisons used the unpaired t-test (normal) or Mann-Whitney U test (non-normal). ns indicating no significance.

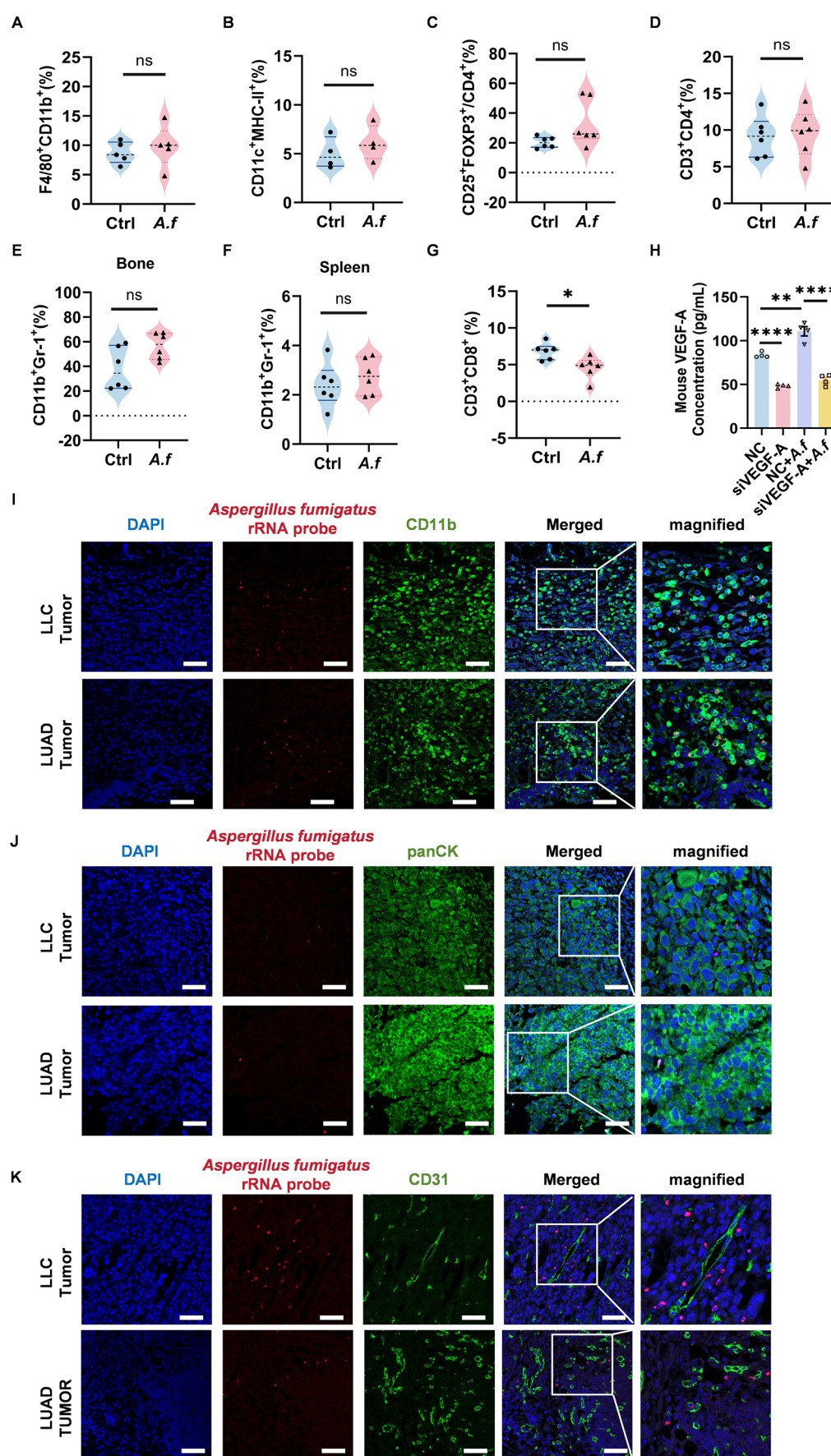

◀

**Figure EV3.** *A. fumigatus* **specifically co-localizes with CD11b⁺ myeloid cells without altering CD4⁺ T cells, macrophages, DCs or Tregs in the tumor microenvironment.**

(A–C) The percentage of macrophages (F4/80⁺ and CD11b⁺ cells) ($p = 0.5584$), dendritic cells (CD11c⁺MHCII⁺) ($p = 0.4231$) and Treg cells (CD25⁺FOXP3⁺CD4⁺) ($p = 0.0772$) in tumor tissues were detected by flow cytometry ($n = 4$–6 biological replicates). (D) Proportions of CD3⁺CD4⁺ T cells ($p = 0.7962$) in tumor tissues were detected by flow cytometry. ($n = 6$ biological replicates) (E, F) Proportions of total MDSCs (CD11b⁺Gr-1⁺) in bone and spleen tissues were detected by flow cytometry (Bone: $p = 0.0506$) (Spleen: $p = 0.4595$) ($n = 5$–6 biological replicates). (G) Proportions of CD3⁺CD8⁺ T cells in tumor tissues were detected by flow cytometry ($p = 0.0154$) ($n = 6$ biological replicates). (H) VEGF-A secretion in the culture supernatant of MDSCs was measured by ELISA. (NC and siVEGF-A: $p < 0.0001$; NC and NC + A. f: $p = 0.0027$; NC + A. f and siVEGF-A + A. f: $p < 0.0001$) ($n = 4$ biological replicates). (I–K) Representative FISH images showing *A. fumigatus* (red) and CD11b⁺ myeloid cells (green), panCK⁺ tumor cells (green), CD31⁺ endothelium (green). Nuclei counterstained with DAPI (blue) (scale bars, 50μm). Data information: Data with error bars are represented as mean ± SEM. *$p < 0.05$, **$p < 0.01$, ****$p < 0.0001$ as determined by unpaired Student's t-test. ns indicating no significance.

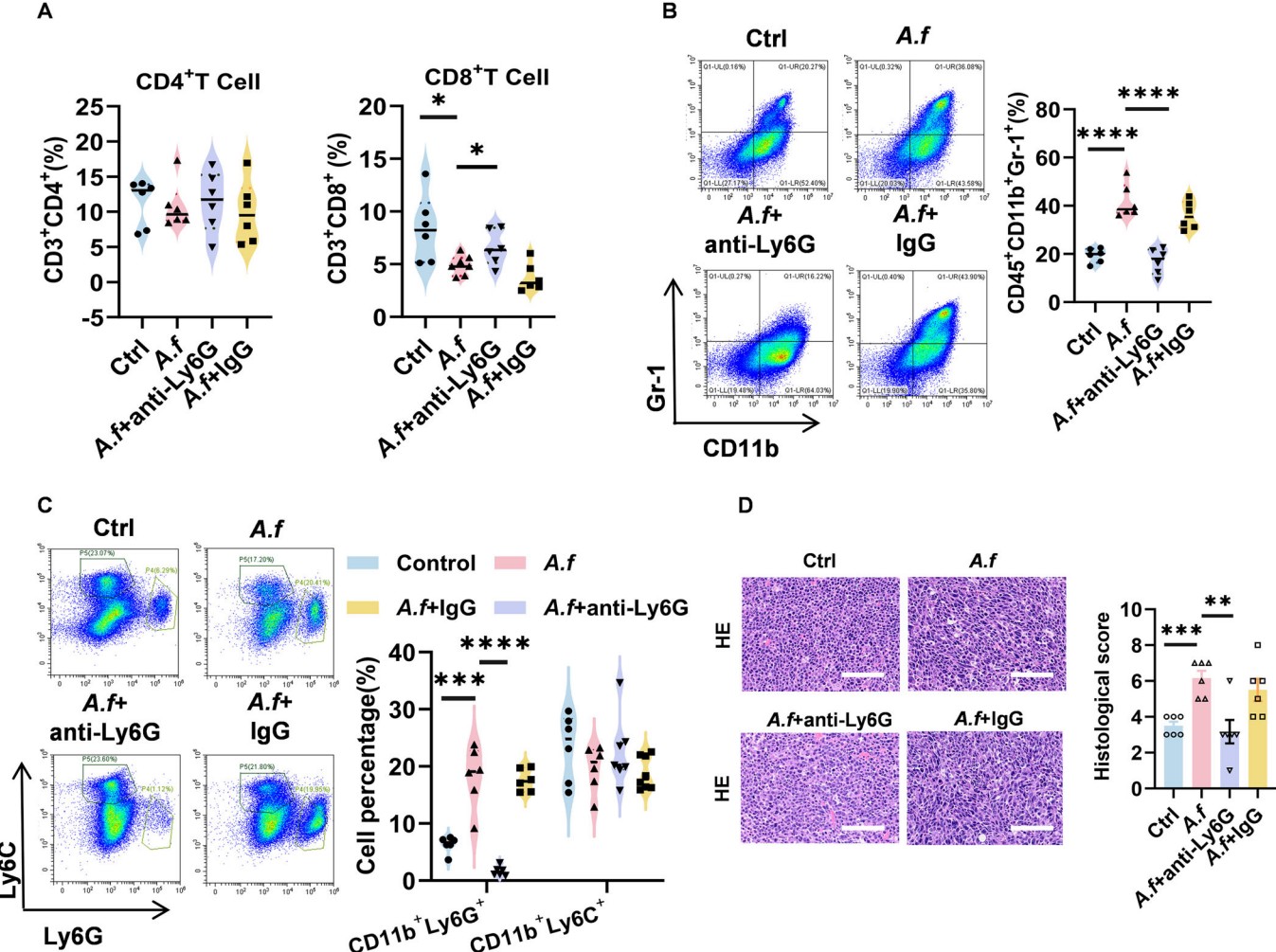

**Figure EV4. G-MDSCs reduction enhances the immune response in tumor-bearing mice treated with peritumor *A. fumigatus* injections.**

(A) Proportions of T cell subgroups (CD3+CD4+ T cells and CD3+CD8+ T cells) in tumor tissues were detected by flow cytometry. (CD8+ T cells: Ctrl and *A. f*: $p = 0.0181$. *A f* and *A. f* + antiLy6G: $p = 0.0439$) ($n = 6$ biological replicates). (B, C) Proportions of total MDSCs (CD45+CD11b+Gr-1+), G-MDSCs (CD11b+Ly6G+) and M-MDSCs (CD11b+Ly6C+) in tumor tissues were detected by flow cytometry (MDSCs: Ctrl and *A. f*: $p < 0.0001$. *A f* and *A. f* + antiLy6G: $p < 0.0001$; G-MDSCs: Ctrl and *A. f*: $p = 0.0003$. *A f* and *A. f* + antiLy6G: $p < 0.0001$) ($n = 6$ biological replicates). (D) HE histological analysis of tumors was shown by HE staining (scale bars, 100 μm) (Ctrl and *A. f*: $p = 0.0002$. *A f* and *A. f* + antiLy6G: $p = 0.0045$) ($n = 6$ biological replicates). Data information: Data with error bars are represented as mean ± SEM. Normality was assessed using the Shapiro-Wilk test. Two-group comparisons used the unpaired t-test (normal) or Mann-Whitney U test (non-normal). *$p < 0.05$, **$p < 0.01$, ***$p < 0.001$, ****$p < 0.0001$ as determined by unpaired Student's t-test or Mann-Whitney U test.

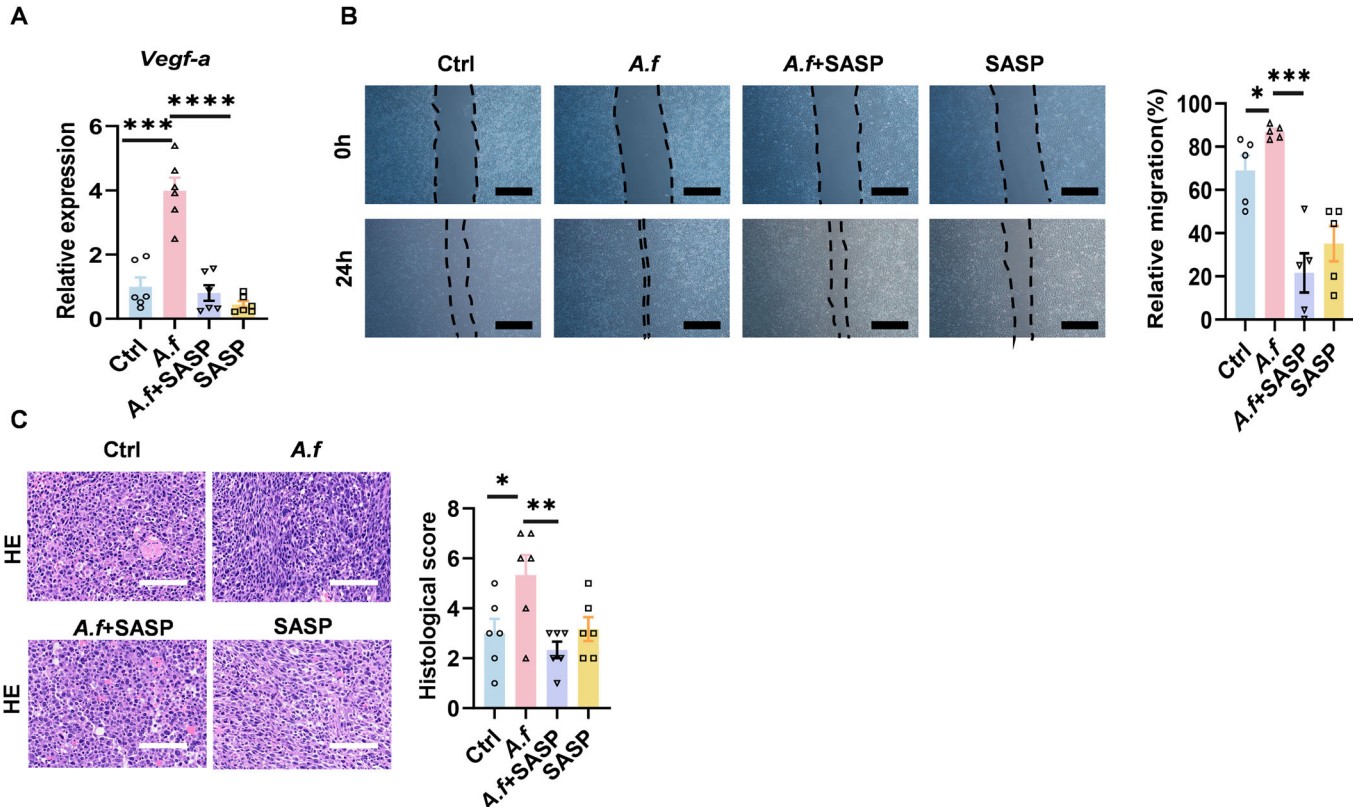

**Figure EV5. SLC7A11 mediates *A. fumigatus*-induced pro-Angiogenic function of MDSCs.**

The inhibitor of slc7a11, sulfasalazine (SASP) (2 μM), was added to the supernatant of the culture medium of MDSCs. Medium supernatants from MDSCs were collected and used as conditioned medium for HUVEC cells. (A) The gene expression of *Vegf-a* in MDSCs was detected by qPCR (Ctrl and *A. f*: $p = 0.0001$; *A. f* and *A. f* + SASP: $p < 0.0001$) ($n = 6$ biological replicates). (B) The migratory ability of HUVEC cells were assessed (scale bars, 500 μm) (Ctrl and *A. f*: $p = 0.0135$; *A. f* and *A. f* + SASP: $p = 0.0001$) ($n = 5$ biological replicates). (C) SASP (100 mg/kg) was injected intraperitoneally daily in the LLC mouse model. HE histological analysis of tumors was shown by HE staining (scale bars, 100 μm) (Ctrl and *A. f*: $p = 0.0400$; *A. f* and *A. f* + SASP: $p = 0.0062$) ($n = 6$ biological replicates). Data information: Data with error bars are represented as mean ± SEM. *$p < 0.05$, **$p < 0.01$, ***$p < 0.001$, ****$p < 0.0001$ as determined by unpaired Student's t-test.

