## [Peer Review File · EMBO Reports]

Aspergillus fumigatus promotes tumor angiogenesis via SLC7A11 on myeloid-derived suppressor cells

Wei Qu, Zelin Wang, Tianchen Zhu, Huiyue Cui, Ziqian Bing, Sunan Shen, Yi Shen, Shaorong Yu, Hongqin Zhuang, and Tingting Wang

Corresponding author(s): Tingting Wang (wangtt@nju.edu.cn) , Hongqin Zhuang (hqzhuang@nju.edu.cn), Yi Shen (dryishen@nju.edu.cn), Shaorong Yu (shaorongyu@njmu.edu.cn)

Review Timeline:

Submission Date:	15th Jan 25
Editorial Decision:	28th Mar 25
Revision Received:	26th Jul 25
Editorial Decision:	26th Aug 25
Revision Received:	1st Oct 25
Accepted:	17th Oct 25

Editor: Achim Breiling

Transaction Report:

Dear Prof. Wang,

Thank you for the submission of your manuscript to EMBO reports. I have now received the reports from the three referees that were asked to evaluate your study, which can be found at the end of this email.

As you will see, the referees think that these findings are of interest. However, they have several comments, concerns, and suggestions, indicating that a major revision of the manuscript is necessary to allow publication of the study in EMBO reports. As the reports are below, and all the referee concerns need to be addressed, I will not detail them here.

Given the constructive referee comments, I would like to invite you to revise your manuscript with the understanding that the concerns of the referees must be addressed in the revised manuscript and in a detailed point-by-point response. Acceptance of your manuscript will depend on a positive outcome of a second round of review. It is EMBO reports policy to allow a single round of revision only and acceptance of the manuscript will therefore depend on the completeness of your responses included in the next, final version of the manuscript.

- 1) a .docx formatted version of the final manuscript text (including legends for main figures, EV figures and tables), but without the figures included. Figure legends should be compiled at the end of the manuscript text.
- 2) individual production quality figure files as .eps, .tif, .jpg (one file per figure), of main figures and EV figures. Please upload these as separate, individual files upon re-submission.

- 4) a complete author checklist, which you can download from our author guidelines (<https://www.embopress.org/page/journal/14693178/authorguide>). Please insert page numbers in the checklist to indicate where the requested information can be found in the manuscript. The completed author checklist will also be part of the RPF.

- 5) that primary datasets produced in this study (e.g. RNA-seq, ChIP-seq, structural and array data) are deposited in an

appropriate public database. If no primary datasets have been deposited, please also state this in a dedicated section (e.g. 'No primary datasets have been generated and deposited'), see below.

The accession numbers and database should be listed in a formal "Data Availability" section that follows the model below. This is now mandatory (like the COI statement). Please note that the Data Availability Section is restricted to new primary data that are part of this study. This section is mandatory. As indicated above, if no primary datasets have been deposited, please state this in this section

Data availability

8) Regarding data quantification and statistics, please make sure that the number "n" for how many independent experiments were performed, their nature (biological versus technical replicates), the bars and error bars (e.g. SEM, SD) and the test used to calculate p-values is indicated in the respective figure legends (also for EV and Appendix figures). Please also check that all the p-values are explained in the legend, and that these fit to those shown in the figure. Please provide statistical testing where applicable. Please avoid the phrase 'independent experiment', but clearly state if these were biological or technical replicates. Please also indicate (e.g. with n.s.) if testing was performed, but the differences are not significant. In case n=2, please show the data as separate datapoints without error bars and statistics. See also: <http://www.embopress.org/page/journal/14693178/authorguide#statisticalanalysis>

9) Please add scale bars of similar style and thickness to microscopic images, using clearly visible black or white bars (depending on the background). Please place these in the lower right corner of the images themselves. Please do not write on or near the bars in the image but define the size in the respective figure legend.

10) Please also note our reference format:

12) We now use CRedit to specify the contributions of each author in the journal submission system. CRedit replaces the author contribution section. Please use the free text box to provide more detailed descriptions and do NOT provide your final manuscript text file with an author contributions section. See also our guide to authors: <https://www.embopress.org/page/journal/14693178/authorguide#authorshipguidelines>

13) All Materials and Methods need to be described in the main text using our 'Structured Methods' format, which is required for

all research articles. According to this format, the Methods section should include a Reagents and Tools Table (listing key reagents, experimental models, software, and relevant equipment and including their sources and relevant identifiers), uploaded as separate file, and a Methods section in which we encourage the authors to describe their methods using a step-by-step protocol format with bullet points, to facilitate the adoption of the methodologies across labs. More information on how to adhere to this format as well as downloadable templates (.doc) for the Reagents and Tools Table can be found in our author guidelines (section 'Structured Methods'):

14) Please order the sections like this, using (only) these names:

Title page - Abstract - Keywords - Introduction - Results - Discussion - Methods - Data availability section - Acknowledgements (including the funding information) - Disclosure and Competing Interests Statement - References - Figure legends - Expanded View Figure legends

15) Please make sure that all the funding information is also entered into the online submission system and that it is complete and similar to the one in the acknowledgement section of the manuscript text file.

Please note that corresponding authors are required to supply an ORCID ID upon submission of a revised manuscript and an institutional e-mail address. Please make sure that all co-corresponding authors provide an ORCID and an institutional e-mail address in the submission system. Please find instructions on how to link the ORCID ID to the account in our manuscript tracking system in our Author guidelines: <http://www.embopress.org/page/journal/14693178/authorguide#authorshipguidelines>

I look forward to seeing a revised form of your manuscript when it is ready.

Yours sincerely,

Referee #1:

1. Appropriate length and format for the type of article submitted YES
2. Physiological/functional relevance demonstrated (detailed insight into the mechanism is not always necessary). The MS is dealing with a complex and fascinating area of interactions of environmental molds (*aspergillus fumigatus*) in the airways with carcinogenesis (lung cancer). There scarcity of literature regarding such interactions, in contrast to the extensive evidence provided for the GI fungal mycobiome (yeasts) and cancer of the gastrointestinal tract
3. Strong evidence for the conclusions that are drawn YES. The MS follows a logical story line and experimental rigor is very good. s are
4. Novelty (abstracts, meeting reports & online preprints do not compromise novelty) YES, and has a simple yet key message that these association between of interactions of environmental molds (*aspergillus fumigatus*) in the airways with carcinogenesis (lung cancer), merits further exploration of such observation/hypothesis and mechanistic insights
5. Broad biological significance YES
6. Importance to the specific field YES

1. An initial paragraph that summarises the major finding and the referee's overall impressions, as well as highlighting any major strengths or shortcomings of the manuscript.

In contrast to microbiome which has become i recognized as modulator tumor angiogenesis)and responses to chemo, check point inhibitors, etc), the role of mycobiome this process remains less known and most data comes from GI mycobiome (yeasts) in GIO cancers. In this study, using in vitro and in vivo experiments, the authors investigated the effects of heat killed *Aspergillus*

fumigatus (*A. fumigatus*) on angiogenesis in lung cancer. Peritumoral *A. fumigatus* worsened tumor burden and increased angiogenesis, in a proangiogenic independent fashion, which is interesting. The authors observed that heat killed *A. fumigatus* enhanced the accumulation of myeloid-derived suppressor cells (in the tumor tissues, which was the catalyst of VEGF-A production and the formation of endothelial lumens. Furthermore, we identified candidates in regulation of the proangiogenic function through MDSCs. These results, although mechanistically preliminary, are novel and invite further studies in that area of how symbiotic fungi in mucosal surfaces, the "tone" host immunity and establishment/progression of tumorigenesis. The MS is well written, with cohesive progression of experiments that have adequate experimental rigor and support the conclusions. It would benefit from some more discussion so they put their findings in a more balanced context

Major comments

- A) The authors used heat kills *Aspergillus*. Is it of any usefulness to use alive *Aspergillus* spores, as fungus is not dead typically in the airways? Is heat killed fungal spores recruitment of MDSCs fungal specific? Using some appropriate controls (e.g. heat killed bacteria) would enhance the specificity of their findings
- B) Of note AF294 *Asp fumigatus* isolate produces gliotoxin, That mycotoxin (in alive *Aspergillus* spores) modulates angiogenesis (see Ben-Ami R et al. *Blood* 2009, *J Infect Dis* 2013). Perhaps the authors should take a look in relevant papers and discuss that point, as limitation of the generalizability of their findings.
- C) There is tremendous strain to strain variation of *Aspergillus* virulence, attachment to endothelial cells, proinflammatory activity, etc. Understandably (using multiple *Aspergillus* strains would make the experiments way too cumbersome). The same can be said for non-*Aspergillus fumigatus* spp (e.g. *Asp favus*) and non-*Aspergillus* molds that colonize airways. Please comment on limitation of using one strain of *Asp fumigatus*
- D) There is an artificiality of such experiments that needs to be recognized further, *Aspergillus* (typically alive spores, not dead spores) interacts in complex way in different sites of respiratory tract (sinus epithelium, upper respiratory airways, Mucus, etc) and distal airways and environmental cues from surrounding immune cells and /or bacteria are complex. For example respiratory bacteria can promote lung adenoCa progression (eg Shen J et al. *Front Microbiol* 2023)
- E) Can you also discuss, beyond the use of one fungal isolate, the generativity if one uses a single cell line and one type of mouse model)
- F) There is significant clinical literature of lung cancer developing in patients with preexisting aspergillomas and *Aspergillus* found in endobronchial tumors (eg Kaplun O et al. *Int J Pathol & Microbiol* 2021, Ham HS et al *Tub & resp diseases* 2006)). Consider expanding on those clinical reports to give more clinical context
- G) Are these findings site (respiratory endothelium) specific? If so, why?
- H) See /expand similar work using *Asp fumigatus* conidia introduction intravenously in a different mouse genetic background with some mechanistic hints (Sohrabi N et al. *Can J Microbiol* 2010)

Minor points:

- I) Abstract: Specify that you used heat killed spores, need abbreviation Of SLC7A11 and HMGB1 as the reader might not know what these acronyms are. How the manuscript might be shortened (including the removal of non-essential experimental data to supplementary information)
- J) Introduction: Specify that GI mycobiome interaction are with yeast (Liu et al, Wang et al, etc) as the term fungal is too general (would make your report more appealing as you deal with molds)
- K) Pg 4; Give more info regarding Angiogenesis (Zhao et al, Lamont et al). Angiogenesis in what cell type? Where?
- L) Sometimes the reader is unclear how if he reads in vitro vs in vivo experiments, consider clarifying
- M) Clarify abbreviations in text: eg LUAD, GSEA, LLC, SLC7A11 help the reader understand more by giving more background (eg Pg 6, what do you mean by Ji67 indicators?)
- N) Pg 8: is it *A. fumigatus* treated or *A. fumigatus*-exposed?
- O) Pg 9: what is the relevance of ferroptosis in MDSCs in these data? Please explain more
- P) Do you have any further supporting histopath for the animal model?

Referee #2:

In their manuscript, Qu and colleagues report an enhanced lung tumor angiogenesis triggered by the fungus, *Aspergillus fumigatus*. Mechanistically, they posit that the angiogenesis is triggered by the action of *A. fumigatus* on g-MDSCs, triggering their expression of the solute carrier, SLC7A11, which binds HMGB1. This is an original and interesting work on the emerging field linking local microbiota (here fungi) to lung tumor development, which could have impact in the clinic. However, there are currently multiple issues. Several comments are written, which hopefully will help the authors.

Major comments:

- 1) Authors used the terminology "gMDSC", while these cells are, in the reviewer's point of view, neutrophils, a term not used in the study. This should be clarified, especially as "MDSC" implies immune suppression, which is never tested here.
- 2) Because only a single model is used in vivo, with tumor cells growing under the skin which is not relevant for lung cancer, authors should validate their findings using at least an orthotopic LUAD model, adding a condition where *A. fumigatus* (or heated

compounds of it) are inhaled, to mimic more adequately the reality of lung microbiome dysbiosis.

3) Neutrophil depletion using anti-Ly6G antibody is challenging, and should be monitored appropriately, as demonstrated in <https://doi.org/10.1038/s41467-020-16596-9>. Here, reviewer is afraid that treating mice only 3x per week will not result in cell depletion. This should be assessed, but not by using anti-Gr1 or anti-Ly6G antibodies as the authors did, the reason being of antigen masking (= apparent but not real reduction in cell numbers). Intracellular Ly6G staining, or staining other specific neutrophil markers should be used instead. Currently the authors cannot use the word "depletion" as this is not demonstrated.

4) Figure 3. Authors transfected monocyte-MDSCs for 48 hrs with siVEGF-A and then performed *A. fumigatus* stimulation for another 24 hrs. Authors should focus on performing experiments on neutrophils since they identified this population as playing a role in *A. fumigatus* induced angiogenesis. It is well known that neutrophils have a very short lifespan, particularly ex-vivo, and siRNA on primary neutrophils are tricky to realise. Could the authors perform this experiment on neutrophils and stain for viability with Annexin-V and 7-AAD after 72 hrs of culture (with or without siRNA)?

5) Authors state that only MDSC population changes upon *A. fumigatus* treatment, whereas they report a significant decrease in CD3+ CD8+ cells (Figure 2D and Figure S4A). Anti-Ly6G antibody did not restore CD8 T cell levels (Figure S4A), suggesting that *A. fumigatus* has an effect on CD8 T cells independently of neutrophils. Can the authors comment? Could this decrease also influence tumor growth and angiogenesis?

6) Figure 6. Overall, data regarding SLC7A11 and HMGB1 interaction are weak and authors are encouraged to strengthen this part. Reviewer wonders, for example, if it would be possible to mutagenise the part of SLC7A11 involved in HMGB1 interaction to demonstrate loss of binding in this situation.

7) Figure 5. SASP increases tumor growth in absence of *A. fumigatus* (5B-D), but this is not mentioned. Please explain.

8) Figure 4D. The experiment on ferroptosis does not currently bring anything to the study. It should be considered to remove this panel.

Minor comments:

1) Figure 1G: Could authors specify what they mean by "Histological score", and add it in the figure legend? G: Tumor differentiation state seems different between ctrl condition and *A. fumigatus* treated condition. Could the authors comment on that? What is exactly meant by "tumor malignancy"?

2) Figure S2. D, G and I: Authors show but do not comment about these figures. General comment: Could authors specify where they did unpaired Student t-test in this figure and found significant differences, as specified in the end of the legend?

3) Figure S3J-K. Authors state that anti-Ly6G treatment decrease angiogenic markers such as CD34 and VEGF-A serum level. However, no significant difference regarding VEGF-A serum level (Figure S3K) was found between the compared group -A.F + anti-Ly6G and A.F + IgG-. Could the author comment on that?

4) Figure 3. To strengthen their results, authors should redo the experiment of Figure 3A (or Figure 3G) adding a condition where supernatant is incubated at RT with anti-VEGF-A antibody (or isotype control) to capture available VEGF-A that was produced by MDSCs and monitor HUVEC tube formation.

5) Figure 4E. It is curious that SASP already increases Vegf-a levels compared to its control condition. Could authors explain this?

6) Statistical tests: there is a discrepancy between what is written in the Material and Methods section, and what is stated in the figure legend. The authors should clearly specify which statistical test was used in the figures and ensure that the appropriate statistical test is applied. The authors should verify the number of experimental datapoint, assess the normality of the samples and consider the number of conditions to compare before selecting the correct statistical test.

7) Figure 5. It is confusing to see "sulfasalazine" and "SASP". Are there the same, and the same as salazosulfapyridine? Please simplify.

8) If the authors are quoting in the text Figure 2C-D and then Figure 2A-B, they should invert them to maintain a logical chronological order.

9) Figure 2. In the legend, authors should remove p-value significance of ** since there is no such data in the figure.

10) Figure 3. In the legend, (B), (E), (I) "and migratory ability" should be removed since authors don't show it in the figure.

11) Legend Figure 5 and 6. Authors should add the p-value of *** stars since they used it in the figure.

12) Supplementary Figure 4. Authors should remove *** p-value in the legend since it is not used in this figure.

13) "cysteine" should be replaced by "cystine"

Referee #3:

In this study, Qu and co-workers found that *Aspergillus fumigatus* infection could promote tumor angiogenesis in lung cancer. The authors showed that the pro-angiogenic effect of *Aspergillus fumigatus* was mainly mediated by MDSCs, rather than acting on tumor cells or vascular endothelial cells. Mechanistically, the authors reveal that SLC7A11 regulated VEGFA secretion by MDSCs, which drove the pro-angiogenic effect. Overall, this is a highly innovative study that proposes a novel role of lung fungi in cancer progression. It reveals how crosstalk between fungi, immune cells, and endothelial cells contributes to lung cancer development through angiogenesis. However, addressing the following questions will significantly improve this manuscript. Specific comments:

1. In Figure 1H, using the TCGA dataset, the authors demonstrated that multiple pathways - including the angiogenesis pathway and FGF pathway - are enriched in *A. fumigatus*-positive samples. Given that the key gene investigated in this study, SLC7A11, plays a critical role in ferroptosis regulation, it would be valuable to explore whether *A. fumigatus* influences ferroptosis by leveraging the TCGA data.
2. Although the study focuses on tumor-infiltrating MDSCs, the authors should clarify whether *A. fumigatus* also modulates MDSC abundance in peripheral compartments (e.g., bone marrow or spleen). This analysis could help determine whether the fungal effects are localized to the tumor microenvironment or systemic.
3. In Figure 4, the authors use a SLC7A11-specific inhibitor to determine whether the pro-angiogenic effect of MDSCs is mediated by SLC7A11 expression. However, MDSCs extracted from bone marrow can't reflect the actual situation in tumors. So, I suggest using MDSCs sorted from the tumor tissue by magnetic bead to redo this experiment.
4. In this article, the author first proposed that fungi can promote the generation of tumor blood vessels. In Introduction, the author mentioned that bacteria also play a role in promoting angiogenesis in tumors, so the author should further discuss the different mechanisms by which bacteria and fungi promote angiogenesis in tumors, as well as possible reasons.
5. SLC7A11, a well-established ferroptosis-related protein, mediates glutamate and cystine transport. However, this study found that SLC7A11 does not appear to modulate ferroptosis to affect tumor progression. The authors should clarify and discuss potential explanations for this observation.
6. In Figure 1H, the authors found that angiogenesis and FGF signaling pathways were highly enriched in *A. fumigatus* positive TCGA tumor samples. But TCGA seems to have no information about the microorganisms inside the tumor. Therefore, it would be helpful if the author could clearly indicate the data source used in Figure 1H in the Methods.

We sincerely thank all reviewers for their valuable time, insightful comments, and constructive suggestions. Addressing this feedback has significantly strengthened our manuscript. Below, we provide detailed point-by-point responses to each reviewer's comments.

Referee #1:

1. An initial paragraph that summarises the major finding and the referee's overall impressions, as well as highlighting any major strengths or shortcomings of the manuscript.

In contrast to microbiome which has become a recognized modulator of tumor angiogenesis (and responses to chemo, check point inhibitors, etc), the role of mycobiome in this process remains less known and most data comes from GI mycobiome (yeasts) in GI cancers. In this study, using in vitro and in vivo experiments, the authors investigated the effects of heat killed *Aspergillus fumigatus* (A. fumigatus) on angiogenesis in lung cancer. Peritumoral A. fumigatus worsened tumor burden and increased angiogenesis, in a proangiogenic independent fashion, which is interesting. The authors observed that heat killed A. fumigatus enhanced the accumulation of myeloid-derived suppressor cells (in the tumor tissues, which was the catalyst of VEGF-A production and the formation of endothelial lumens. Furthermore, we identified candidates in regulation of the proangiogenic function through MDSCs. These results, although mechanistically preliminary, are novel and invite further studies in that area of how symbiotic fungi in mucosal surfaces, the "tone" of host immunity and establishment/progression of tumorigenesis. The MS is well written, with cohesive progression of experiments that have adequate experimental rigor and support the conclusions. It would benefit from some more discussion so they put their findings in a more balanced context.

Major comments

A) The authors used heat-killed *Aspergillus*. Is it of any usefulness to use live *Aspergillus* spores, as fungus is not dead typically in the airways? Is heat-killed fungal spore recruitment of MDSCs fungal specific? Using some appropriate controls (e.g. heat-killed bacteria) would enhance the specificity of their findings.

Response: We sincerely appreciate the reviewer's insightful suggestion. Considering that fungi are not typically present in the airways, we agree with the reviewer that the usage of live *Aspergillus* spores has greater clinical value. Therefore, we incorporated two inhalation models using live *Aspergillus fumigatus* spores in our revised manuscript (revised Figure 1A-O).

Recruitment of MDSCs by heat-killed fungal spores is not fungus-specific. Qiming Zhou et al. demonstrated that bacterial translocation can be mediated by activation of the S100A11-RAGE axis in colonic adenomas, which facilitates immune escape by recruiting myeloid-derived suppressor cells (Zhou et al, 2025). Besides, to investigate the fungal-specific activation of MDSCs' pro-angiogenic capacity, we established an in vitro co-culture system. We collected supernatants of MDSCs in the co-culture system for testing their effect on the lumen-forming ability of endothelial cells. And we also detected *Vegf-a* gene expression of MDSCs in the co-culture system. We found that the supernatant of MDSCs co-cultured with heat-inactivated *Aspergillus fumigatus* was able to induce endothelial cell lumen formation most significantly.

Moreover, heat-killed *Aspergillus fumigatus* was able to most significantly upregulate *Vegf-a* gene expression in MDSCs.

Considering the reviewer's comments, heat killed bacteria need to be used as a suitable control. LPS (lipopolysaccharide) is a major component of the outer membrane of the cell wall of Gram-negative bacteria. According to the reference, LPS can be used to represent heat-inactivated bacteria (Maldonado *et al*, 2016). The results of lumen formation assay and qPCR assay showed that although LPS also had pro-angiogenic ability in the co-culture system, it was not as significant as that of *Aspergillus fumigatus*. This result suggests that the *Aspergillus*-induced enhancement of the proangiogenic capacity of MDSCs was the most significant.

Figure for referee with unpublished data and its description has been removed upon request by the authors.

B) Of note AF294 *Asp fumigatus* isolate produces gliotoxin, That mycotoxin (in alive *Aspergillus* spores) modulates angiogenesis (see Ben-Ami R *et al* .BLLOD 2009, J Infect Dis 2013). Perhaps the authors should take a look in relevant papers and discuss that point, as limitation of the generalizability of their findings.

Response: We thank the reviewer for this valuable comment. We have noted the study by Ben-Ami R *et al* (Ben-Ami *et al*, 2013; Ben-Ami *et al*, 2009), which reported that *Aspergillus fumigatus* and its metabolite gliotoxin can inhibit angiogenesis. Ben-Ami R *et al*. found that the culture filtrate of *Aspergillus fumigatus* had an inhibitory effect on angiogenesis in *in vitro*. It is understandable that the experimental conclusions of Ben-Ami *et al* differ from our experimental

results. In our study, we co-cultured *Aspergillus fumigatus* directly with endothelial cells and confirmed that *Aspergillus fumigatus* did not have a significant effect on endothelial cell migration, proliferation, and angiogenesis. We believe that compared to metabolites, the effect of *Aspergillus fumigatus* itself on endothelial cells is not negligible.

In addition, the study by Ben-Ami R et al. found that gliotoxin in the culture filtrate of *Aspergillus fumigatus* inhibited angiogenesis in a dose-dependent manner. We suggest that in invasive pulmonary aspergillosis, angiogenesis is inhibited because of direct damage to endothelial cells by the high load of *Aspergillus fumigatus*. In our study, we found that *Aspergillus fumigatus* exerts its proangiogenic effect by promoting VEGF-A secretion from MDSCs in lung adenocarcinoma model. These suggest that there may be differences in the effects on angiogenesis depending on the disease model and the amount of *Aspergillus fumigatus*.

We have added a discussion on the different effects of *Aspergillus fumigatus* on angiogenesis in the paragraph 2 of Discussion

C) There is tremendous strain to strain variation of *Aspergillus* virulence, attachment to endothelial cells, proinflammatory activity, etc. Understandably (using multiple *Aspergillus* strains would make the experiments way too cumbersome). The same can be said for non-*Aspergillus fumigatus* spp (e.g. *Asp favus*) and non-*Aspergillus* molds that colonize airways. Please comment on limitation of using one strain of *Asp fumigatus*.

Response: We gratefully acknowledge the reviewer's suggestion. As the reviewer pointed out, different strains of *Aspergillus* have different characteristics of virulence, attachment to endothelial cells, proinflammatory activity, etc. We agree with the reviewer that the use of a single strain has limitations. Here, we use *Aspergillus fumigatus* because it has been widely reported in lung cancer cases (Ali *et al*, 2014; Kaplun *et al*, 2021). We admit that there are limitations in the use of one strain. And we will compare the effects of different species of *Aspergillus* moulds on angiogenesis in future studies. We evaluated this limitation in the paragraph 4 of Discussion.

D) There is an artificiality of such experiments that needs to be recognized further, *Aspergillus* (typically alive spores, not dead spores) interacts in complex way in different sites of respiratory tract (sinus epithelium, upper respiratory airways, Mucus, etc) and distal airways and environmental cues from surrounding immune cells and /or bacteria are complex. For example respiratory bacteria can promotes lung adenoCa progression (eg Shen J et al. Front Micribiol 2023)

Response: We sincerely thank the reviewer for their insightful comments. To reduce artificiality and simulate the complex interactions of *Aspergillus* in different parts of the respiratory tract, we added two inhalation models using live *Aspergillus fumigatus* spores (revised Figure 1A-O). In addition, we also found that *Aspergillus fumigatus* primarily colonized the lung parenchyma and tumor tissue by the newly added FISH experiment (revised Figure 1H) (revised Figure 1O). Although spores could be detected in the bronchial cavity, their number was significantly lower in bronchial cavity than those in the lung parenchyma and tumor tissue. Therefore, in this study, we

focused on the role of *Aspergillus fumigatus* in the tumor in the following study.

E) Can you also discuss, beyond the use of one fungal isolate, the generativity if one uses a single cell line and one type of mouse model

Response: We appreciate the reviewer's comments. Since there is no cell line for MDSCs, we used the extraction method of MDSCs from bone marrow derived primary cells induced by IL-6 and GM-CSF (Bancaro *et al*, 2023). To enhance the generalizability of our findings, we used the human lung adenocarcinoma cell line SPCA1 and the mouse lung adenocarcinoma cell line LLC. We demonstrated that *A. fumigatus* had no effect on migration, proliferation, and pro-angiogenic gene expression of SPCA1 and LLC (revised Figure EV2M-R). In addition, we have added inhalation animal experiments in orthotopic lung cancer model and the subcutaneous tumor model of lung cancer (revised Figure 1A-O) (revised Figure 2B-J). These results show that *Aspergillus fumigatus* significantly promotes lung cancer tumor progression and angiogenesis in mice.

F) There is significant clinical literature of lung cancer developing in patients with preexisting aspergillomas and *Aspergillus* found in endobronchial tumors (eg Kaplun O *et al*. *Ind J Pathol & Microbiol* 2021, ham HS *et al* *Tub & resp diseases* 2006)). Consider expanding on those clinical reports to give more clinical context

Response: Thank you for providing this important suggestion and sharing relevant literature. In order to provide a more comprehensive clinical background, we have added these clinical reports on the coexistence of *aspergillosis* in lung cancer patients in the paragraph 1 of Discussion, emphasizing the clinical correlation between aspergillosis infection and lung cancer. Moreover, we collected samples from clinical patients and confirmed the presence of *Aspergillus fumigatus* in the tumour tissues of lung cancer patients by fluorescence in situ hybridization experiments (revised Figure EV2B).

G) Are these findings site (respiratory endothelium) specific? If so, why?

Response: We appreciate the constructive comment of the reviewer. We found that *Aspergillus fumigatus* did not co-localize with the endothelial cell marker CD31 in tumor tissues by FISH experiments. *Aspergillus fumigatus* also did not co-localize with the tumor cell marker panCK in tumor tissues. *Aspergillus fumigatus* mainly co-localized with the myeloid cell marker CD11b in tumor tissues (revised Figure EV3I-K).

We consider *Aspergillus fumigatus* induces angiogenesis primarily by activating MDSCs to secrete VEGF-A, rather than acting directly on endothelial cells.

H) See /expand similar work using *Asp fumigatus* conidia introduction intravenously in a different mouse genetic background with some mechanistic hints (Sohrabi N *et al*. *Can J Microbiol* 2010)

Response: Thank you to the reviewer for reminding us of the existence of this important reference

(Sohrabi *et al*, 2010). They reported that intravenous *Aspergillus fumigatus* conidia to BALB/c mice promoted mammary tumor growth through systemic Treg expansion and TIMP-1-upregulation. The authors used tail vein injection of *Aspergillus fumigatus* spores in a subcutaneous tumour mouse model of breast cancer. However, we used the method of allowing mice to inhale *Aspergillus fumigatus* or peritumourally inject *Aspergillus fumigatus* in mouse model of lung cancer. It is suggested that *Aspergillus fumigatus* locally and systemically may regulate tumour progression through different immune mechanisms. We cited this article as supplemental information regarding different mouse genetic backgrounds and additional mechanistic insights **in the paragraph 3 of Discussion**.

Minor points:

I) Abstract: Specify that you used heat killed spores, need abbreviation Of SLC7A11m HMGB1 as the reader might not know what these acronyms are. How the manuscript might be shortened (including the removal of non-essential experimental data to supplementary information)

Response: Thank you for very careful review and meaningful comments. Considering that we have referred to your comments (major comments A, D and E), we used a mouse subcutaneous lung cancer tumor model and an orthotopic lung cancer model. And we have demonstrated that inhalation of live *A. fumigatus* spores promotes tumor progression and angiogenesis (**revised Figure 1A-O**) (**revised Figure 2B-J**). Furthermore, we demonstrated that both peritumoral injection of heat-inactivated *A. fumigatus* spores (**Figure 1**) and live *A. fumigatus* spores (**revised Figure EV1C-I**) (**revised Figure 4F-L**) promoted tumor progression and angiogenesis. Therefore, we did not emphasize heat inactivation in our abstract.

We have added the full name of SLC7A11 as well as HMGB1 in order to make it easier for the reader to understand it.

To shorten the manuscript and to take into account that ferroptosis was not altered in MDSCs, we removed data related to ferroptosis (**revised Figure 4**). We put some negative results in the supplementary data. For example, *Aspergillus fumigatus* had no effect on changes in the proportion of immune cells, such as macrophages, in the tumor microenvironment. We put this part of the flow-through results into the supplemental data (**revised Figure EV3A-D**).

J) Introduction: Specify that GI mycobiome interaction are with yeast (Liu et al, wang et al, etc) as the term fungal is too general (would make your report more appealing as you deal with molds).

Response: Thank you for your valuable suggestion. As recommended, we have revised the **Introduction** by replacing the general term "fungi" with "yeast" to more accurately reflect that the gastrointestinal mycobiome interactions discussed (e.g., in Liu et al., Wang et al.).

K) Pg 4; Give more info regarding Angiogenesis (Zhao et al, Lamont et al). Angiogenesis in what cell type? Where?

Response: Thank you for pointing out this problem in manuscript. Regarding the angiogenesis

information on page 4: Zhao et al. demonstrated that colonization with *P. micra* derived from colorectal cancer patients upregulates pro-angiogenic gene expression in situ within colon tissues of mice.

Lamont et al. further elucidated that this pro-angiogenic effect is exerted primarily on endothelial cells.

These specific details on location and cell types have now been added to the manuscript.

K) Sometimes the reader is unclear how if he reads in vitro vs in vivo experiments, consider clarifying

Response: Thank you for your suggestion. We have revised the manuscript to clearly distinguish between in vivo and in vitro experiments. We have added animal model diagrams (revised Figure 3A) and cell model diagrams (revised Figure 3F) in the Results section to help distinguish between in vivo and in vitro experiments. Furthermore, we have added explicit textual indications within the Results section.

M) Clarify abbreviations in text: eg LUAD, GSEA, LLC, SLC7A11 help the reader understand more by giving more background (eg Pg 6, what do you mean by Ki67 indicators?)

Response: Thank you so much for your careful check. Abbreviation Clarification: As recommended, we have defined all abbreviations (LUAD, GSEA, LLC, SLC7A11) at their first occurrence in the text with their full terms: Lung Adenocarcinoma (LUAD); Gene Set Enrichment Analysis (GSEA); Lewis Lung Carcinoma (LLC); Solute Carrier Family 7 Member 11 (SLC7A11).

We have clarified that Ki67 staining in tumor tissues primarily serves as an indicator of tumor cell proliferation activity in manuscript.

N) Pg 8: is it *A. fumigatus* treated or *A. fumigatus*-exposed?

Response: Thank you for the reviewer's reminder. To ensure clarity and uniformity throughout the manuscript, we have carefully revised the text and replaced all instances of "*A. fumigatus*-exposed" with "*A. fumigatus* treated".

O) Pg 9: what is the relevance of ferroptosis in MDSCs in these data? Please explain more

Response: We studied ferroptosis in MDSCs because SLC7A11 is a key regulatory gene in the ferroptosis process. However, considering that ferroptosis is not altered in MDSCs, we have removed the data related to ferroptosis (revised Figure 4).

P) Do you have any further supporting histopath for the animal model?

Response: Thank you for your suggestion. In order to provide more histopath support, we added pro-angiogenic factor VEGF-A indicator in all animal models (revised Figure 2C) (revised Figure 2I) (revised Figure 4H) (revised Figure 5M). Immunohistochemistry results showed that VEGF-A-positive staining was significantly enhanced in the tumor tissues of the *Aspergillus fumigatus* group.

Referee #2:

In their manuscript, Qu and colleagues report an enhanced lung tumor angiogenesis triggered by the fungus, *Aspergillus fumigatus*. Mechanistically, they posit that the angiogenesis is triggered by the action of *A. fumigatus* on g-MDSCs, triggering their expression of the solute carrier, SLC7A11, which binds HMGB1. This is an original and interesting work on the emerging field linking local microbiota (here fungi) to lung tumor development, which could have impact in the clinic. However, there are currently multiple issues. Several comments are written, which hopefully will help the authors.

Major comments:

1) Authors used the terminology "gMDSC", while these cells are, in the reviewer's point of view, neutrophils, a term not used in the study. This should be clarified, especially as "MDSC" implies immune Suppression, which is never tested here.

Response: We appreciate the reviewer's insightful comments regarding the use of gMDSCs and neutrophil terminology. The use of the terminology "gMDSCs" versus "neutrophils" is indeed significantly controversial. Because myeloid-derived suppressor cells (MDSCs) are pathologically activated neutrophils and monocytes with potent immunosuppressive activity (Veglia *et al*, 2021).

We also agree with the reviewer that MDSCs imply immunosuppression. After we referred to the reviewer's review comment (5), we modified our protocol to reduce g-MDSCs. Using the modified protocol, CD8 T cells were partially rebounded (revised Figure EV4A). This experimental result suggests that this population of g-MDSCs is immunosuppressive.

Considering this result, we still use the terminology "gMDSCs".

2) Because only a single model is used in vivo, with tumor cells growing under the skin which is not relevant for lung cancer, authors should validate their findings using at least an orthotopic LUAD model, adding a condition where *A. fumigatus* (or heated compounds of it) are inhaled, to mimic more adequately the reality of lung microbiome dysbiosis.2)

Response: We thank the reviewer for these constructive suggestions. In order to more accurately model the imbalance of the lung microbiota, we supplemented our experimental results with the inhalation of live *Aspergillus fumigatus* spores in the orthotopic model of lung cancer (revised Figure 1I-O). And we also supplemented the experiments with inhalation of live *Aspergillus*

fumigatus spores in subcutaneous tumour mouse model of lung cancer (revised Figure 1A-H). These results showed that inhalation of live *Aspergillus fumigatus* spores resulted in accelerated tumour progression and increased angiogenesis in the lung cancer model mice (revised Figure 2B-J).

3) Neutrophil depletion using anti-Ly6G antibody is challenging, and should be monitored appropriately, as demonstrated in <https://doi.org/10.1038/s41467-020-16596-9>. Here, reviewer is afraid that treating mice only 3x per week will not result in cell depletion. This should be assessed, but not by using anti-Gr1 or anti-Ly6G antibodies as the authors did, the reason being of antigen masking (= apparent but not real reduction in cell numbers). Intracellular Ly6G staining, or staining other specific neutrophil markers should be used instead. Currently the authors cannot use the word "depletion" as this is not demonstrated.

Response: We thank the reviewers for their valuable comments. According to this reference, we repeated the experiment of reducing g-MDSCs by using the "Combo" protocol(Boivin *et al*, 2020)(revised Figure 4A). And we confirmed the significant reduction of g-MDSCs by intracellular staining (revised Figure EV4C). In addition, since we were able to detect g-MDSCs by flow cytometry, we changed "deletion" to "reduction" in the manuscript.

4) Figure 3. Authors transfected monocyte-MDSCs for 48 hrs with siVEGF-A and then performed *A. fumigatus* stimulation for another 24 hrs. Authors should focus on performing experiments on neutrophils since they identified this population as playing a role in *A. fumigatus* induced angiogenesis. It is well known that neutrophils have a very short lifespan, particularly ex-vivo, and siRNA on primary neutrophils are tricky to realise. Could the authors perform this experiment on neutrophils and stain for viability with Annexin-V and 7-AAD after 72 hrs of culture (with or without siRNA)?

Response: We agree with the reviewer that neutrophils have a very short lifespan in vitro culture. Thank you also for pointing out the problem of possible cell viability of primary cells cultured in vitro. Our method of extracting bone marrow-derived primary MDSCs was referred to the method of references(Bancaró *et al.*, 2023; Qin *et al*, 2023). The method of using siRNA to treat MDSCs is in reference to Vasquez-Dunddel *et al* (Tan *et al*, 2025; Vasquez-Dunddel *et al*, 2013). And we also verified that bone marrow-derived MDSCs remained more than 88% viable after 72 hours by the method of staining Annexin-V and 7-AAD. With these results we confirmed that bone marrow-derived MDSCs can maintain viability in vitro.

Figure for referee with unpublished data and its description has been removed upon request by the authors.

5) Authors state that only MDSC population changes upon *A. fumigatus* treatment, whereas they report a significant decrease in CD3⁺ CD8⁺ cells (Figure 2D and Figure S4A). Anti-Ly6G antibody did not restore CD8 T cell levels (Figure S4A), suggesting that *A. fumigatus* has an effect on CD8 T cells independently of neutrophils. Can the authors comment? Could this decrease also influence tumor growth and angiogenesis?

Response: We thank the reviewer for their important suggestions. Considering that *Aspergillus fumigatus* treatment leads to changes in the number of CD3⁺CD8⁺ cells in the tumor microenvironment as well. To test whether *Aspergillus fumigatus* affects angiogenesis via CD8 T cells, we co-cultured *Aspergillus fumigatus* with CD8 T cells. We collected supernatants from CD8 T cells co-cultured with *Aspergillus fumigatus* and examined the effect of the supernatants on endothelial cell lumen formation. In addition, we tested whether *Aspergillus fumigatus* affected the expression of pro-angiogenic genes in CD8 T cells. The results of lumen formation as well as qPCR experiments indicated that *Aspergillus fumigatus* could not affect angiogenesis via CD8 T cells.

Figure for referee with unpublished data and its description has been removed upon request by the authors.

In addition, the number of CD8 T cells was partially restored after we referred to the method of removing g-MDSCs proposed by the reviewer in comment (5). Therefore, we suggest that the reduction of CD8 T cells in the tumor microenvironment results from the immunosuppressive function of g-MDSCs (revised Figure EV4A). Furthermore, we add a discussion about the effect of changes CD8 T cell numbers on tumor growth and angiogenesis in paragraph4 of the Discussion section. We suggest that *Aspergillus fumigatus* leads to an increase in g-MDSCs in tumors. The accelerated tumor progression and increased tumor angiogenesis caused by *Aspergillus fumigatus* was significantly suppressed after removal of g-MDSCs. However, the number of CD8 T cells was partially restored. These experimental results suggest that *Aspergillus fumigatus* affects tumor progression mainly through the pro-angiogenic function of MDSCs.

6) Figure 6. Overall, data regarding SLC7A11 and HMGB1 interaction are weak and authors are encouraged to strengthen this part. Reviewer wonders, for example, if it would be possible to mutagenise the part of SLC7A11 involved in HMGB1 interaction to demonstrate loss of binding in this situation

Response: We sincerely thank the reviewer for this valuable suggestion to strengthen the evidence for the SLC7A11-HMGB1 interaction. We performed site-directed mutagenesis on SLC7A11, targeting residues predicted by ZDOCK to be involved in binding. Specifically, we constructed plasmids harboring point mutations and assessed their impact on HMGB1 binding using co-immunoprecipitation assays. Our results demonstrate that mutations at Asp386, Leu299, and Gln71 on SLC7A11 significantly weakened its interaction with HMGB1 (revised Figure 6I). These new data provide crucial experimental validation identifying key binding sites and substantially strengthen the evidence for this specific molecular interaction within our proposed mechanism.

7) Figure 5. SASP increases tumor growth in absence of *A. fumigatus* (5B-D), but this is not mentioned. Please explain.

Response: We thank the reviewers for pointing out this problem. We considered that although SASP is an antitumour drug, it has poor bioavailability (Bagherpoor *et al*, 2023). Gavage administration is now more often reported to be used to alleviate the inflammatory response in current studies. We referred to a literature (Mao *et al*, 2021) and corrected the administration method of SASP, changing it from gavage to intraperitoneal injection. Given the above literature, we repeated the SASP animal administration experiment. After changing the administration method, SASP did not cause tumor growth (revised Figure 5G).

8) Figure 4D. The experiment on ferroptosis does not currently bring anything to the study. It

should be considered to remove this panel.

Response: Thank you for the reviewer's comments. We investigated ferroptosis in MDSCs because SLC7A11 is a key regulatory gene during ferroptosis. However, considering that ferroptosis was not altered in MDSCs, we removed the data related to ferroptosis.

Minor comments:

1) Figure 1G: Could authors specify what they mean by "Histological score", and add it in the figure legend? G: Tumor differentiation state seems different between ctrl condition and A. fumigatus treated condition. Could the authors comment on that? What is exactly meant by "tumor malignancy"?

Response: Thank you so much for your careful check. Histological scoring mainly evaluating from the aspects of "Tumor Cell Morphology", "Tumor Necrosis", "Infiltration of inflammatory cells", and "invasion". We have also added specific scoring methods to the histological scoring criteria in the Appendix Table3.

We note that *Aspergillus fumigatus* may affect the differentiation status of tumours. To verify whether it is the increased angiogenesis that leads to dedifferentiation, we would also need to administer anti-vascular therapeutic drugs concurrently in a mouse model of *Aspergillus fumigatus* inhalation and then test the extent of tumour differentiation. We have added a discussion of the differences in the differentiation status of tumours in paragraph4 of the Discussion section.

In addition, we sincerely appreciate the reviewer's insightful comment. We have replaced "tumor malignancy" with "tumor progression" throughout the manuscript. This terminology better aligns with our mechanistic findings.

2) Figure S2. D, G and I: Authors show but do not comment about these figures. General comment: Could authors specify where they did unpaired Student t-test in this figure and found significant differences, as specified in the end of the legend?

Response: We thank the reviewer for their insightful suggestions. We modified the presentation of the results in Figure S2. D, G and I. We have also revised the presentation of the findings in the revised Figure 2. Figure S2D is to demonstrate that *Aspergillus fumigatus* had no statistically significant differential effect on angiogenic gene expression in the A549 cell line (revised Figure 2K). Figure S2G is to demonstrate that *Aspergillus fumigatus* had no statistically significant differential effect on proliferation of the HUVEC cell line (revised Figure 2C). Figure S2I shows that *Aspergillus fumigatus* had no statistically significant differential effect on angiogenic gene expression in the HUVEC cell line (revised Figure 2D).

We have systematically re-evaluated all analyses to ensure correct test application, with necessary recalculations performed. Normality was assessed for each comparison group using the Shapiro-Wilk test. Based on this assessment, two-group comparisons used the unpaired t-test

(normal) or Mann-Whitney U test (non-normal) for the Ctrl and *A.f* groups. ns indicating no significance.

3) Figure S3J-K. Authors state that anti-Ly6G treatment decrease angiogenic markers such as CD34 and VEGF-A serum level. However, no significant difference regarding VEGF-A serum level (Figure S3K) was found between the compared group -A.F + anti-Ly6G and A.F + IgG-. Could the author comment on that?

Response: Thank you for pointing out this problem in manuscript. We repeated the anti-Ly6G treatment experiment with reference to the reviewer's comment (5). And we increased the number of samples. We found that serum VEGF-A levels were significantly lower in the A.F+ anti-Ly6G group compared to the A.F+ IgG group (revised Figure 4I), consistent with the CD34 findings.

4) Figure 3. To strengthen their results, authors should redo the experiment of Figure 3A (or Figure 3G) adding a condition where supernatant is incubated at RT with anti-VEGF-A antibody (or isotype control) to capture available VEGF-A that was produced by MDSCs and monitor HUVEC tube formation.

Response: Thank you for the above suggestions. We collected supernatants from MDSCs co-incubated with anti-VEGF-A antibody. The effect of the supernatant on the lumen-forming ability of endothelial cells was assayed. The results showed that the *Aspergillus fumigatus*-induced increase in lumen formation of the MDSCs supernatants could be significantly inhibited by the VEGF-A antibody (revised Figure 3L). This experimental result strengthens our conclusion.

5) Figure 4E. It is curious that SASP already increases Vegf-a levels compared to its control condition. Could authors explain this?

Response: We sincerely thank the reviewer for careful reading. We repeat the experiment of Figure 4E. We found that the elevated *Vegf-a* mRNA expression in the SASP group relative to the control group was not reproducible (revised Figure EV5A). This result aligns with the absence of significant differences in VEGF-A protein expression between the SASP group and the control group (revised Figure 5E).

6) Statistical tests: there is a discrepancy between what is written in the Material and Methods section, and what is stated in the figure legend. The authors should clearly specify which statistical test was used in the figures and ensure that the appropriate statistical test is applied. The authors should verify the number of experimental datapoint, assess the normality of the samples and consider the number of conditions to compare before selecting the correct statistical test.

Response: We thank the reviewer for identifying the statistical reporting discrepancy and their valuable suggestions. We have revised the Materials and Methods section to detail our statistical selection criteria. We verified the number of biological replicates (n = 4-6 per group for most comparisons), clearly stating them in figures/legends. Normality was assessed for each

comparison group using the Shapiro-Wilk test. Based on this assessment, two-group comparisons used the unpaired t-test (normal) or Mann-Whitney U test (non-normal). For tumor growth curves, involving repeated measures, we employed two-way ANOVA analysis. We have systematically re-evaluated all analyses to ensure correct test application, with necessary recalculations performed.

7) Figure 5. It is confusing to see "sulfasalazine" and "SASP". Are there the same, and the same as salazosulfapyridine? Please simplify.

Response: Sorry for the confusing caused to the reviewer. SASP refers to sulfasalazine. We have corrected all spelling errors in the manuscript.

8) If the authors are quoting in the text Figure 2C-D and then Figure 2A-B, they should invert them to maintain a logical chronological order.

Response: We sincerely appreciate the reviewer's meticulous review of our manuscript. We have made corresponding modifications to the order of the figures in revised Figure 3B-C and EV3A-D.

9) Figure 2. In the legend, authors should remove p-value significance of ** since there is no such data in the figure.

Response: Thank you for the reviewer's reminder. We have modified the relevant content in the legend in revised Figure 3 and EV3.

10) Figure 3. In the legend, (B), (E), (I) "and migratory ability" should be removed since authors don't show it in the figure.

Response: We thank the reviewer for their thorough examination. We have removed "and migratory ability" in the legend of revised Figure 3.

11) Legend Figure 5 and 6. Authors should add the p-value of *** stars since they used it in the figure.

Response: Thank you for the reviewer's correction. We have modified the relevant content in the legend in revised Figure 5 and 6.

12) Supplementary Figure 4. Authors should remove *** p-value in the legend since it is not used in this figure.

Response: Thank you for the reviewer's reminder. We have modified the relevant content in the legend in revised Figure EV4.

13) "cysteine" should be replaced by "cystine"

Response: Thank you for the reviewer's correction. We have made the corresponding modifications in manuscript.

Referee #3:

In this study, Qu and co-workers found that *Aspergillus fumigatus* infection could promote tumor angiogenesis in lung cancer. The authors showed that the pro-angiogenic effect of *Aspergillus fumigatus* was mainly mediated by MDSCs, rather than acting on tumor cells or vascular endothelial cells. Mechanistically, the authors reveal that SLC7A11 regulated VEGFA secretion by MDSCs, which drove the pro-angiogenic effect. Overall, this is a highly innovative study that proposes a novel role of lung fungi in cancer progression. It reveals how crosstalk between fungi, immune cells, and endothelial cells contributes to lung cancer development through angiogenesis. However, addressing the following questions will significantly improve this manuscript.

Specific comments:

1. In Figure 1H, using the TCGA dataset, the authors demonstrated that multiple pathways - including the angiogenesis pathway and FGF pathway - are enriched in *A. fumigatus*-positive samples. Given that the key gene investigated in this study, SLC7A11, plays a critical role in ferroptosis regulation, it would be valuable to explore whether *A. fumigatus* influences ferroptosis by leveraging the TCGA data.

Response: Thank you for the above suggestions. We analyzed ferroptosis-related gene expression in *Aspergillus fumigatus*-positive samples from TCGA dataset. Only SLC7A11 was found to be significantly upregulated. This result is consistent with our experimental findings that *Aspergillus fumigatus* did not ferroptosis in MDSCs.

Figure for referee with unpublished data and its description has been removed upon request by the authors.

2. Although the study focuses on tumor-infiltrating MDSCs, the authors should clarify whether *A. fumigatus* also modulates MDSC abundance in peripheral compartments (e.g., bone marrow or

spleen). This analysis could help determine whether the fungal effects are localized to the tumor microenvironment or systemic.

Response: We gratefully thanks for the precious time the reviewer spent making constructive remarks. We analyzed changes in the number of MDSCs in bone marrow sites as well as in splenic sites (revised Figure EV3E-F). We found no significant changes in the number of MDSCs in the bone marrow and spleen. We considered that the role of fungal is limited to the tumor microenvironment.

3. In Figure 4, the authors use a SLC7A11-specific inhibitor to determine whether the pro-angiogenic effect of MDSCs is mediated by SLC7A11 expression. However, MDSCs extracted from bone marrow can't reflect the actual situation in tumors. So, I suggest using MDSCs sorted from the tumor tissue by magnetic bead to redo this experiment.

Response: We totally understand the reviewer's concern. We found in vitro that *Aspergillus fumigatus* was able to induce significant upregulation of the *Slc7a11* gene in bone marrow-derived MDSCs. To increase the persuasiveness of this finding. We added in vivo validation in an animal model. Using magnetic beads, we isolated G-MDSCs from tumor tissues. we found that with the inhibition of in vitro experiments, the *Slc7a11* gene was significantly upregulated in MDSCs from tumor tissues of mice in the *Aspergillus fumigatus* group.

Figure for referee with unpublished data and its description has been removed upon request by the authors.

4. In this article, the author first proposed that fungi can promote the generation of tumor blood vessels. In Introduction, the author mentioned that bacteria also play a role in promoting angiogenesis in tumors, so the author should further discuss the different mechanisms by which bacteria and fungi promote angiogenesis in tumors, as well as possible reasons.

Response: Thank you for the above suggestions. We suggest that metabolites, cell wall components may be important factors in the different possible mechanisms regarding angiogenesis between fungi and bacteria. These differences may lead to different cells in the tumor

microenvironment to respond to microbes. And it would activate different signaling pathways in the cells. We have added content about the differences in the mechanisms by which fungi and bacteria influence tumor angiogenesis in paragraph 4 of the Discussion section.

5. SLC7A11, a well-established ferroptosis-related protein, mediates glutamate and cystine transport. However, this study found that SLC7A11 does not appear to modulate ferroptosis to affect tumor progression. The authors should clarify and discuss potential explanations for this observation.

Response: Thank for your comments. SLC7A11 is now widely reported to be associated with iron death, but in our study, we found that although *A. fumigatus* induced significant up-regulation of Slc7a11 in MDSCs, iron death-related metrics were not significantly altered in MDSCs. We believe that our study adds about the non-classical function of this protein, SLC7A11. SLC7A11 can also regulate tumor angiogenesis in MDSCs through interaction with HMGB1.

6. In Figure 1H, the authors found that angiogenesis and FGF signaling pathways were highly enriched in *A. fumigatus* positive TCGA tumor samples. But TCGA seems to have no information about the microorganisms inside the tumor. Therefore, it would be helpful if the author could clearly indicate the data source used in Figure 1H in the Methods.

Response: Thank you for pointing out this problem in manuscript. Based on their results (Narunsky-Haziza et al. *Cell*. 2022), we identified the pathways highly enriched in *A. fumigatus*-positive samples by GSEA with a threshold of adjusted P-value < 0.05. We found that gene expression of angiogenesis-related pathways was significantly upregulated. We specify the data sources used in revised Figure 2A.

References

- Ali S, Malik A, Bhargava R, Shahid M, Fatima N (2014) Aspergillus colonization in patients with bronchogenic carcinoma. *Asian Cardiovasc Thorac Ann* 22: 460-464
- Bagherpoor AJ, Shameem M, Luo X, Seelig D, Kassie F (2023) Inhibition of lung adenocarcinoma by combinations of sulfasalazine (SAS) and disulfiram-copper (DSF-Cu) in cell line models and mice. *Carcinogenesis* 44: 291-303
- Bancaro N, Cali B, Troiani M, Elia AR, Arzola RA, Attanasio G, Lai P, Crespo M, Gurel B, Pereira R *et al* (2023) Apolipoprotein E induces pathogenic senescent-like myeloid cells in prostate cancer. *Cancer Cell* 41: 602-619.e611
- Ben-Ami R, Albert ND, Lewis RE, Kontoyiannis DP (2013) Proangiogenic growth factors potentiate in situ angiogenesis and enhance antifungal drug activity in murine invasive aspergillosis. *J Infect Dis* 207: 1066-1074
- Ben-Ami R, Lewis RE, Leventakos K, Kontoyiannis DP (2009) Aspergillus fumigatus inhibits angiogenesis through the production of gliotoxin and other secondary metabolites. *Blood* 114: 5393-5399
- Boivin G, Faget J, Ancey PB, Gkasti A, Mussard J, Engblom C, Pfirschke C, Contat C, Pascual J, Vazquez J *et al* (2020) Durable and controlled depletion of neutrophils in mice. *Nat Commun* 11: 2762
- Kaplun O, Papamanoli A, Chernyavskiy I, Pseudos G (2021) Endobronchial carcinoid tumor coexisting with saprophytic Aspergillus. *Indian J Pathol Microbiol* 64: 413-414
- Maldonado RF, Sá-Correia I, Valvano MA (2016) Lipopolysaccharide modification in Gram-negative bacteria during chronic infection. *FEMS Microbiol Rev* 40: 480-493
- Mao C, Liu X, Zhang Y, Lei G, Yan Y, Lee H, Koppula P, Wu S, Zhuang L, Fang B *et al* (2021)

DHODH-mediated ferroptosis defence is a targetable vulnerability in cancer. *Nature* 593: 586-590

Qin G, Liu S, Liu J, Hu H, Yang L, Zhao Q, Li C, Zhang B, Zhang Y (2023) Overcoming resistance to immunotherapy by targeting GPR84 in myeloid-derived suppressor cells. *Signal Transduct Target Ther* 8: 164

Sohrabi N, Hassan ZM, Khosravi AR, Tebianian M, Mahdavi M, Tootian Z, Ebrahimi SM, Yadegari MH, Gheflati Z (2010) Invasive aspergillosis promotes tumor growth and severity in a tumor-bearing mouse model. *Can J Microbiol* 56: 771-776

Tan L, Kong W, Zhou K, Wang S, Liang J, Hou Y, Dou H (2025) FoxO1 Deficiency in Monocytic Myeloid-Derived Suppressor Cells Exacerbates B Cell Dysfunction in Systemic Lupus Erythematosus. *Arthritis Rheumatol* 77: 423-438

Vasquez-Dunddel D, Pan F, Zeng Q, Gorbounov M, Albesiano E, Fu J, Blosser RL, Tam AJ, Bruno T, Zhang H *et al* (2013) STAT3 regulates arginase-I in myeloid-derived suppressor cells from cancer patients. *J Clin Invest* 123: 1580-1589

Veglia F, Sanseviero E, Gabrilovich DI (2021) Myeloid-derived suppressor cells in the era of increasing myeloid cell diversity. *Nat Rev Immunol* 21: 485-498

Zhou Q, Lei L, Cheng J, Chen J, Du Y, Zhang X, Li Q, Li C, Deng H, Wong CC *et al* (2025) Microbiota-induced S100A11-RAGE axis underlies immune evasion in right-sided colon adenomas and is a therapeutic target to boost anti-PD1 efficacy. *Gut* 74: 214-228

Dear Prof. Wang

Thank you for the submission of your revised manuscript to our editorial offices. I have now received the reports from the two referees that I asked to re-evaluate the study, you will find below. Original referee #3 was completely unresponsive to my invitations to re-assess the study. However, going through your p-b-p-response, I consider her/his concerns as adequately addressed.

As you will see, the remaining referees now support the publication of your study in EMBO reports. However, referee #2 has a remaining important concern I ask you to address experimentally in a final revised manuscript. Please also provide a final p-b-p-response to these points and my editorial requests below.

Editorial requests:

- Please provide the abstract written in present tense throughout.

- We now use CRediT to specify the contributions of each author in the journal submission system. CRediT replaces the author contribution section. Please use the free text box to provide more detailed descriptions and do NOT provide your final manuscript text file with an author contributions section. See also our guide to authors: <https://www.embopress.org/page/journal/14693178/authorguide#authorshipguidelines>

- Then, please order the manuscript sections like this, using only these names:

Title page - Abstract - Keywords - Introduction - Results - Discussion - Methods - Data availability section - Acknowledgements (please put here all the funding information) - Disclosure and Competing Interests Statement - References - Figure legends - Expanded View Figure legends

- Please check again that the number "n" for how many independent experiments were performed, their nature (biological versus technical replicates), the bars and error bars (e.g. SEM, SD) and the test used to calculate p-values is indicated in the respective figure legends. Please also check that all the p-values are explained in the legend, and that these fit to those shown in the figure. Please provide statistical testing where applicable. Please avoid the phrase 'independent experiment' but clearly state if these were biological or technical replicates. Please also indicate (e.g. with n.s.) if testing was performed, but the differences are not significant. In case n=2, please show the data as separate datapoints without error bars and statistics. See also:

<http://www.embopress.org/page/journal/14693178/authorguide#statisticalanalysis>

If n<5, please show single datapoints for diagrams. Moreover:

- Please note that the exact p values are not provided in the legends of figures 1B, D, F, G, K, M, N; 2B-J; 3B, C, D, E, G, H, I, J, K, L; 4B, D, F, G, H, I, J, K, L; 5D, E, G, I, K, L, M, N, O, P, Q; 6D, E, F, G; EV1 D, F, H, I; EV3 G, H; EV4 A-D; EV5 A-C.

- Please note that the box plots need to be defined in terms of minima, maxima, centre, bounds of box and whiskers, and percentile in the legends of figures EV2 H.

- Please note that information related to n is missing in the legend of figure 5B.

- Please add to each legend (main, EV figures and Appendix Figures, where applicable) a 'Data Information' section explaining the statistics used or providing information regarding replicates and scales. See:

- Please make sure all figure panels (main and EV figures) are called out separately and sequentially. Presently, there seem to be no callouts for Appendix Table S4. Please check. Moreover, please change the callouts for the Appendix tables to 'Appendix Table Sx' (the S is presently missing). Finally, please add 'Appendix' (and the S) to the callout for 'Table 2'.

- Please make sure that all the funding information is also entered into the online submission system and that it is complete and similar to the one in the acknowledgement section of the manuscript text file. Presently, the grant '82302934' is missing in the submission system. Please check.

- It seems patient samples have been used in this study. Thus, please fill in the respective fields in the author checklist.

- Moreover, please note that corresponding authors are required to supply an ORCID ID upon submission of a revised manuscript and an institutional e-mail address. Please provide ORCID IDs for the three co-corresponding authors and an institutional e-mail address for co-corresponding author Shaorong Yu. Please find instructions on how to link the ORCID ID to the account in our manuscript tracking system in our Author guidelines:

<http://www.embopress.org/page/journal/14693178/authorguide#authorshipguidelines>

In addition, I would need from you uploaded separately:

Best,

Referee #1:

The authors did a wonderful job addressing in a comprehensive way nearly all the reviewers comments. The manuscript is now much improved, more complete and balance and provides useful information. I recommend approval as it is

Referee #2:

Reviewer thanks the authors for carefully addressing the points raised before and for all the work that has been done.

Reviewer would, however, still highlight a major issue in the current manuscript (major points 1 and 4). Indeed, the authors clearly showed that in vivo, among MDSC cells (CD11b+ GR1+), the neutrophil population (identified as CD11b+ Ly6G+) but not monocyte population (identified as CD11b+ Ly6C+) is driving tumor growth following Aspergillus infection. Moreover, the authors induced a specific depletion of neutrophils using the «Combo» protocol, which again confirmed neutrophil implication in this model. Therefore, the in vitro experiments should be performed using Ly6G+ magnetic beads (e.g., Miltenyi, #130-120-337) instead of GR1+ magnetic beads, to study specifically the cells reported to be important in vivo. Otherwise not the same population of cells is studied throughout the paper.

We sincerely thank the Editor and the Reviewers for their time, insightful comments, and constructive suggestions. Their valuable feedback has greatly improved the quality and clarity of our manuscript. We have revised the manuscript accordingly and provide our point-by-point responses below.

Point by point response to editorial comments:

- Please provide the abstract written in present tense throughout.

Response: Thank you for your suggestion. We have revised the abstract to use the present tense, with the modified sections highlighted in blue.

- We now use CRediT to specify the contributions of each author in the journal submission system. CRediT replaces the author contribution section. Please use the free text box to provide more detailed descriptions and do NOT provide your final manuscript text file with an author contributions section. See also our guide to authors:

<https://www.embopress.org/page/journal/14693178/authorguide#authorshipguidelines>

Response: Thank you for this guidance. We have now used the journal's CRediT system within the submission portal to specify each author's contributions. As instructed, we have also removed the 'Author Contributions' section from the manuscript text file.

- Then, please order the manuscript sections like this, using only these names:

Title page - Abstract - Keywords - Introduction - Results - Discussion - Methods - Data availability section - Acknowledgements (please put here all the funding information) - Disclosure and Competing Interests Statement - References - Figure legends - Expanded View Figure legends

Response: Thank you for the guidance. We have now reorganized the manuscript strictly according to the specified order: Title page, Abstract, Keywords, Introduction, Results, Discussion, Methods, Data availability, Acknowledgements (which now includes all funding information), Disclosure and Competing Interests Statement, References, Figure legends, and Expanded View Figure legends.

- Please check again that the number "n" for how many independent experiments were performed, their nature (biological versus technical replicates), the bars and error bars (e.g. SEM, SD) and the test used to calculate p-values is indicated in the respective figure legends. Please also check that all the p-values are explained in the legend, and that these fit to those shown in the figure. Please provide statistical testing where applicable. Please avoid the phrase 'independent experiment' but clearly state if these were biological or technical replicates. Please also indicate (e.g. with n.s.) if testing was performed, but the differences are not significant. In case n=2, please show the data as separate datapoints without error bars and statistics. See also:

<http://www.embopress.org/page/journal/14693178/authorguide#statisticalanalysis>

Response: Thank you for providing these important statistical guidelines. We have thoroughly

reviewed and revised all figure legends to ensure full compliance. The specific changes are as follows. We have clearly stated the number of replicates (n) and explicitly defined their nature as either biological or technical replicates, avoiding the term "independent experiment." For each graph, we now specify that the error bars represent the mean \pm SEM and indicate the specific statistical test used (e.g., unpaired two-tailed t-test, one-way ANOVA). All p-values are now explained in the figure legends and correspond precisely to the annotations shown in the figures. Non-significant results following statistical testing are indicated with "n.s." Consistent with the guidelines, for any dataset where n=2, the data are presented as individual data points without error bars or statistical testing. These necessary details have now been added to the respective figure legends.

If n<5, please show single datapoints for diagrams. Moreover:

- Please note that the exact p values are not provided in the legends of figures 1B, D, F, G, K, M, N; 2B-J; 3B, C, D, E, G, H, I, J, K, L; 4B, D, F, G, H, I, J, K, L; 5D, E, G, I, K, L, M, N, O, P, Q; 6D, E, F, G; EV1 D, F, H, I; EV3 G, H; EV4 A-D; EV5 A-C.

Response: Thank you for your careful review and for pointing out the lack of exact p-values in the legends of the specified figures. We have now updated the legends for all figures listed (Figures 1B, D, F, G, K, M, N; 2B-J; 3B, C, D, E, G, H, I, J, K, L; 4B, D, F, G, H, I, J, K, L; 5D, E, G, I, K, L, M, N, O, P, Q; 6D, E, F, G; EV1 D, F, H, I; EV3 G, H; EV4 A-D; EV5 A-C) to include the exact p-values. Furthermore, we confirm that in all diagrams where n<5, individual data points are now displayed as requested.

- Please note that the box plots need to be defined in terms of minima, maxima, centre, bounds of box and whiskers, and percentile in the legends of figures EV2 H.

Response: We have labeled the minimum, maximum, centre, box boundaries, whisker boundaries, and percentiles in Figure EV2 H.

- Please note that information related to n is missing in the legend of figure 5B.

Response: We have added information related to the sample size n in the legend of Figure 5B.

- Please add to each legend (main, EV figures and Appendix Figures, where applicable) a 'Data Information' section explaining the statistics used or providing information regarding replicates and scales. See:

Response: We have added a 'Data Information' section to the legend of every figure, including main and expanded view figures. This section explains the statistical tests applied and scale information for each experiment. The nature of replicates, including the number of biological and technical replicates, is also specified in the figure legends. The specific statistical methods, including normality testing with Shapiro-Wilk, followed by unpaired t test, Mann-Whitney U test, or two-way ANOVA for repeated measures, and the use of GraphPad Prism software version 8.0

are now detailed in each respective legend.

- Please make sure all figure panels (main and EV figures) are called out separately and sequentially. Presently, there seem to be no callouts for Appendix Table S4. Please check. Moreover, please change the callouts for the Appendix tables to 'Appendix Table Sx' (the S is presently missing). Finally, please add 'Appendix' (and the S) to the callout for 'Table 2'.

Response: Thank you for your comments. All figure panels in the main and expanded view (EV) figures have now been individually and sequentially referenced throughout the text. We have also added the missing reference to Appendix Table S4 in the Methods section. Furthermore, all calls to appendix tables have been standardized to the 'Appendix Table Sx' format, and 'Table 2' has been updated to 'Appendix Table S2' to ensure consistency.

- Please make sure that all the funding information is also entered into the online submission system and that it is complete and similar to the one in the acknowledgement section of the manuscript text file. Presently, the grant '82302934' is missing in the submission system. Please check.

Response: We have verified that the funding information has been completely entered into the online submission system and is consistent with the content in the manuscript text file.

- It seems patient samples have been used in this study. Thus, please fill in the respective fields in the author checklist.

Response: We have completed the respective sections regarding patient samples in the author checklist.

- Moreover, please note that corresponding authors are required to supply an ORCID ID upon submission of a revised manuscript and an institutional e-mail address. Please provide ORCID IDs for the three co-corresponding authors and an institutional e-mail address for co-corresponding author Shaorong Yu. Please find instructions on how to link the ORCID ID to the account in our manuscript tracking system in our Author guidelines:

<http://www.embopress.org/page/journal/14693178/authorguide#authorshipguidelines>

Response: We have now supplied the ORCID IDs for all co-corresponding authors and updated the institutional email address for Dr. Shaorong Yu in the submission system.

- a short, two-sentence summary of the manuscript (not more than 35 words).

Response: We have provided a two-sentence summary (under 35 words) in the designated field of the submission system.

- two to four short (!) bullet points highlighting the key findings of your study (two lines each).

Response: The key findings have been summarized in the required bullet-point format and uploaded.

- a schematic summary figure as separate file that provides a sketch of the major findings (not a data image) in jpeg or tiff format (with the exact width of 550 pixels and a height of not more than 400 pixels) that can be used as a visual synopsis on our website.

Response: A schematic summary figure, prepared according to the specified dimensions (550x400 pixels), has been uploaded as a separate file.

Point by point response to reviewers' comments:

Referee #1:

The authors did a wonderful job addressing in a comprehensive way nearly all the reviewers comments. The manuscript is now much improved, more complete and balance and provides useful information. I recommend approval as it is.

Response: We are sincerely grateful to the reviewer for their positive assessment of our revisions and for recommending approval of our manuscript. We greatly appreciate the time and effort they dedicated to reviewing our work.

Referee #2:

Reviewer thanks the authors for carefully addressing the points raised before and for all the work that has been done.

Reviewer would, however, still highlight a major issue in the current manuscript (major points 1 and 4). Indeed, the authors clearly showed that *in vivo*, among MDSC cells (CD11b⁺ GR1⁺), the neutrophil population (identified as CD11b⁺ Ly6G⁺) but not monocyte population (identified as CD11b⁺ Ly6C⁺) is driving tumor growth following *Aspergillus* infection. Moreover, the authors induced a specific depletion of neutrophils using the «Combo» protocol, which again confirmed neutrophil implication in this model. Therefore, the *in vitro* experiments should be performed using Ly6G⁺ magnetic beads (e.g., Miltenyi, #130-120-337) instead of GR1⁺ magnetic beads, to study specifically the cells reported to be important *in vivo*. Otherwise not the same population of cells is studied throughout the paper.

Response: We sincerely appreciate the reviewer's valuable and highly constructive suggestions. Following the reviewer's suggestions, we used Ly6G⁺ magnetic beads (Miltenyi, #130-120-337) to isolate Ly6G⁺ cells from tumor tissues and cultured them *in vitro* for 48 hours.

We observed that the Ly6G⁺ cells from the *Aspergillus fumigatus* group promoted endothelial tube formation more than controls (revised Figure 3D). Additionally, Ly6G⁺ cells from the *Aspergillus fumigatus* group secreted higher levels of VEGF-A, as measured in the culture supernatant (revised Figure 3E).

In conclusion, we found that *Aspergillus fumigatus* can enhance the pro-angiogenic capacity of Ly6G⁺ cells in tumor tissues and increase their VEGF-A secretion. These findings confirm that the Ly6G⁺ population is the key driver of the pro-angiogenic effects we observed *in vivo* with the "Combo" protocol.

Prof. Tingting Wang
Nanjing University
22 Hankou Road
nanjing 210093
China

Dear Prof. Wang,

I am very pleased to accept your manuscript for publication in the next available issue of EMBO reports. Thank you for your contribution to our journal.

Yours sincerely,
